# Tailoring *d*-band center of high-valent metal-oxo species for pollutant removal via complete polymerization

Hong-Zhi Liu[1,4], Xiao-Xuan Shu[1,4], Mingjie Huang [1,2] ✉, Bing-Bing Wu[3], Jie-Jie Chen [1] ✉, Xi-Sheng Wang [3], Hui-Lin Li[1] & Han-Qing Yu [1] ✉

Polymerization-driven removal of pollutants in advanced oxidation processes (AOPs) offers a sustainable way for the simultaneous achievement of contamination abatement and resource recovery, supporting a low-carbon water purification approach. However, regulating such a process remains a great challenge due to the insufficient microscopic understanding of electronic structure-dependent reaction mechanisms. Herein, this work probes the origin of catalytic pollutant polymerization using a series of transition metal (Cu, Ni, Co, and Fe) single-atom catalysts and identifies the *d*-band center of active site as the key driver for polymerization transfer of pollutants. The high-valent metal-oxo species, produced via peroxymonosulfate activation, are found to trigger the pollutant removal via polymerization transfer. Phenoxyl radicals, identified by the innovative spin-trapping and quenching approaches, act as the key intermediate in the polymerization reactions. More importantly, the oxidation capacity of high-valent metal-oxo species can be facilely tuned by regulating their binding strength for peroxymonosulfate through *d*-band center modulation. A 100% polymerization transfer ratio is achieved by lowering the *d*-band center. This work presents a paradigm to dynamically modulate the electronic structure of high-valent metal-oxo species and optimize pollutant removal from wastewater via polymerization.

The "Water–Energy–Sanitation" crisis is evident considering that ~800 million people live without clean water, 1.1 billion individuals lack access to electricity, and 2.5 billion do not have adequate sanitation[1–6]. In pursuit of delivering potable and industrially safe water, the advanced oxidation process (AOP) has been increasingly adopted as supplement to the conventional biological process to eliminate the prevailing organic pollutants[7,8]. The AOPs with a capacity to produce reactive species (e.g., $SO_4^{\bullet-}$ and $^\bullet OH$) to mineralize a vast array of pollutants are termed M-AOPs. Despite their advantages, M-AOPs have some drawbacks, such as forming more stable and toxic intermediates,

the increase in treatment time and costs, and more importantly, the increase in carbon emissions[7,9–12]. Typically, in M-AOPs, carbons in organic contaminants are eventually converted into $CO_2$, which is then released into the atmosphere. This process results in the inability to recover the substantial chemical energy contained within wastewater[4]. Therefore, to achieve sustainable wastewater treatment and promote resource recovery simultaneously, the conventional M-AOPs are in urgent need of a paradigm shift to simultaneously realize contamination abatement and resource recovery in a low-carbon manner.

[1]CAS Key Laboratory of Urban Pollutant Conversion, Department of Environmental Science and Engineering, University of Science and Technology of China, Hefei, China. [2]School of Environmental Science and Engineering, Huazhong University of Science and Technology, Wuhan, China. [3]Department of Chemistry, University of Science and Technology of China, Hefei, China. [4]These authors contributed equally: Hong-Zhi Liu, Xiao-Xuan Shu. ✉e-mail: mingjiehuang@hust.edu.cn; chenjiej@ustc.edu.cn; hqyu@ustc.edu.cn

Recently, regulating the conversion mechanism of pollutants from mineralization (M-AOPs) to polymerization (P-AOPs) in AOPs has drawn increasing attention[13–17]. P-AOPs aim to convert organic pollutants from aqueous solutions into recyclable products to solid surfaces via polymerization transfer (PT) reactions. Thus, they can simultaneously achieve energy conservation and emission reduction for water purification. Previously, the PT processes were predominantly reported in the metal oxides and carbonaceous materials-based catalytic systems by using persulfates or $H_2O_2$ as the oxidant with a varied PT ratio (Eq. 1, the feature is the synchronous removal of total organic carbon (TOC) and pollutants)[17–20]. However, achieving a highly efficient pollutant removal via PT is still a great challenge due to the very limited understanding about the structure–function relationship between the electronic structure of the catalyst and the PT ratio.

$$PT\ ratio = \frac{TOC\ removal\ efficiency}{pollutant\ removal\ efficiency} \times 100\% \qquad (1)$$

To elucidate the structure-dependent catalytic features in heterogeneous AOPs, single-atom catalysts (SACs) are the widely selected platform due to their explicit and flexible catalytic structures[21]. Currently, extensive research on SACs-based AOPs for pollutant removal has been conducted[22–26]. However, a widely overlooked inconsistency existed in the reported SACs-based AOPs. That is, the actual oxidant consumption was lower than the theoretical oxidant consumption for pollutant removal in Cu-SACs[27] and Co-SACs[26], indicative of a typical polymerization reaction feature. However, such a phenomenon did not exist in Fe-SACs[28] (Supplementary Table 1). Such a sharp divergence implies that the metal species might dynamically change the mineralization/polymerization removal pathways of pollutants. However, the fundamental understanding about how the PT process is initialized and what feature of the catalyst governs the PT ratio is still elusive. This poses a great obstacle in delivering a general guiding principle for designing catalytic systems with nearly 100% selectivity for pollutant removal via PT.

Herein, we present a systematic in-depth investigation into peroxymonosulfate (PMS)-based heterogeneous catalytic AOPs over a series of transition metal (TM: Cu, Ni, Co, and Fe)-SACs and reveal the underlying mechanism for achieving a 100% PT ratio by modulating the $d$-band center of the catalytic center. Initially, the similar physicochemical properties and identical $TM–N_4$ active sites of the four TM-SA/PN-g-$C_3N_4$ catalysts were confirmed with a suite of characterizations. Subsequently, an innovative method was established to identify phenoxyl radicals in AOPs. Thereafter, the critical roles of high-valent metal-oxo species, e.g., Cu(III)–OH, Ni(IV)=O, Co(IV)=O, and Fe(IV)=O, in the polymerization of pollutants were elucidated. Last, the $d$-band center-dependent mechanism for achieving 100% PT ratio was revealed with a combination of theoretical calculations and experimental results. Overall, this work unravels a mechanism for pollutant removal in the SACs-AOPs and proposes a foundational guideline for designing SACs-based catalytic systems with 100% selectivity for pollutant removal via the robust PT process.

## Results
### Synthesis and characterizations of TM-SA/PN-g-$C_3N_4$
A hydrogen-bonding-assisted pyrolysis strategy was employed to fabricate a series of single transition metal-embedded pyrrolic N-rich graphitic carbon nitride (TM-SA/PN-g-$C_3N_4$, M = Cu, Ni, Co, Fe, etc.) catalysts. This facile method allowed for the gram-scale production of TM-SACs with high metal loadings (Supplementary Methods and Fig. 1a)[29]. Inductively coupled plasma atomic emission spectroscopy (ICP-AES) analysis indicates that the metal loading was 9.46, 7.02, 7.71, and 7.32 wt% for Cu, Ni, Co, and Fe, respectively (Supplementary Fig. 1a). The absence of characteristic crystal peaks of metals in the powder X-ray diffraction (XRD) patterns of the TM-SA/PN-g-$C_3N_4$

samples (Supplementary Fig. 1b) excluded the presence of large crystalline particles[30,31]. Likewise, high-resolution transmission electron microscopy (HR-TEM) did not detect any visible nanoparticles (Fig. 1b and Supplementary Fig. 2). Scanning electron microscopy (SEM) images reveal that the TM-SA/PN-g-$C_3N_4$ samples possessed similar curved flake-like morphology with a porous structure. The main mesopore structure endorsed the TM-SA/PN-g-$C_3N_4$ with high surface areas as indicated by the Brunauer–Emmett–Teller (BET) measurements (Fig. 1b and Supplementary Figs. 3 and 4). Energy-dispersive spectroscopy (EDS) mapping images verify the uniform distributions of the metal, C, and N elements across these architectures (Fig. 1c and Supplementary Fig. 5). Then, aberration-corrected high-angle annular dark-field scanning transmission electron microscopy (AC HAADF-STEM) with sub-angstrom resolution was utilized to identify the dispersion of metal species[32]. As shown in Fig. 1d and Supplementary Fig. 6, high-density single metal atoms represented as isolated bright dots marked with red circles were uniformly dispersed in the TM-SA/PN-g-$C_3N_4$ samples. Such observations could be further confirmed by the three-dimensional (3D) isoline and atomic overlapping Gaussian function fitting mappings (Fig. 1e and Supplementary Fig. 6). Furthermore, the intensity distribution along $X–Y$ in reveals the atomic spacing for Cu, Ni, Co, and Fe-SACs were approximately 0.34, 0.36, 0.35, and 0.35 nm, respectively (Fig. 1d and Supplementary Fig. 6). Overall, these results confirm the atomic loading of Cu, Ni, Co, and Fe on the TM-SA/PN-g-$C_3N_4$ samples without the existence of metal-derived crystalline structures.

The electronic and atomic structures of metal atoms in the TM-SA/PN-g-$C_3N_4$ samples were further explored using X-ray absorption fine structure (XAFS) spectra. Taking Co-SA/PN-g-$C_3N_4$ as an example, the X-ray absorption near edge structure (XANES) profiles in Fig. 1f shows that the energy absorption intensity of the white line for Co-SA/PN-g-$C_3N_4$ was higher than that for Co foil and lower than that for CoO, indicating that the valence state of the Co atom lay between $Co^0$ and $Co^{2+}$. The lower valence state of Co than its precursor signifies a notable metal-support interaction within the Co-SA/PN-g-$C_3N_4$ sample. Besides, Fourier transform extended X-ray absorption fine structure (FT-EXAFS) spectra of $R$-space for the Co-SA/PN-g-$C_3N_4$ exhibited a principal peak at about 1.5 Å, deviating from the metal scattering path of Co–Co at about 2.2 Å in Co foil. This result confirms the isolated distribution of Co atoms on the PN-g-$C_3N_4$ support (Fig. 1g). Consistent with the FT-EXAFS analysis, wavelet transform (WT) contour plots visually show only the intensity maximum related to Co–N or Co–O interaction, without any Co–Co interaction for the Co-SA/PN-g-$C_3N_4$ (Fig. 1h).

Furthermore, the quantitative coordination configuration of Co in the Co-SA/PN-g-$C_3N_4$ was analyzed by EXAFS fitting. The simulation indicates a primary peak stemming from Co–N or Co–O first shell coordination with a calculated coordination number of approximately 4.0 (Supplementary Figs. 9 and 10). The optimal-fitting results from the EXAFS data are concisely detailed in Supplementary Table 2. Nevertheless, due to the potential presence of O atoms in the microenvironment, the precise coordination structure of the Co single atom remained unclear. Therefore, time-of-flight secondary ion mass spectrometry (TOF-SIMS) measurement using $Bi^{3+}$ ion beam sputtering was utilized to validate the coordination structure of Co in the Co-SA/PN-g-$C_3N_4$ by probing the molecular weight ($M_n$) of the Co-$X_4$ (X = N or O) unit (Supplementary Fig. 11a)[25]. Typical secondary negative and positive ions ($m/z^- = 113.99$, $m/z^+ = 116.03$) were detected in the Co-SA/PN-g-$C_3N_4$, affirming the atomic Co–$N_4$ site in the Co-SA/PN-g-$C_3N_4$ (Fig. 1i). Such $TM–N_4$ structure was also verified for the other TM-SA/PN-g-$C_3N_4$ (TM = Cu, Ni, and Fe) samples with the similar analysis methods (Supplementary Figs. 7–11 and Supplementary Note 1). Moreover, the C and N $K$-edge XANES spectra confirm that metal doping had minimal impact on the atomic structure of the PN-g-$C_3N_4$ substrate (Supplementary Fig. 12).

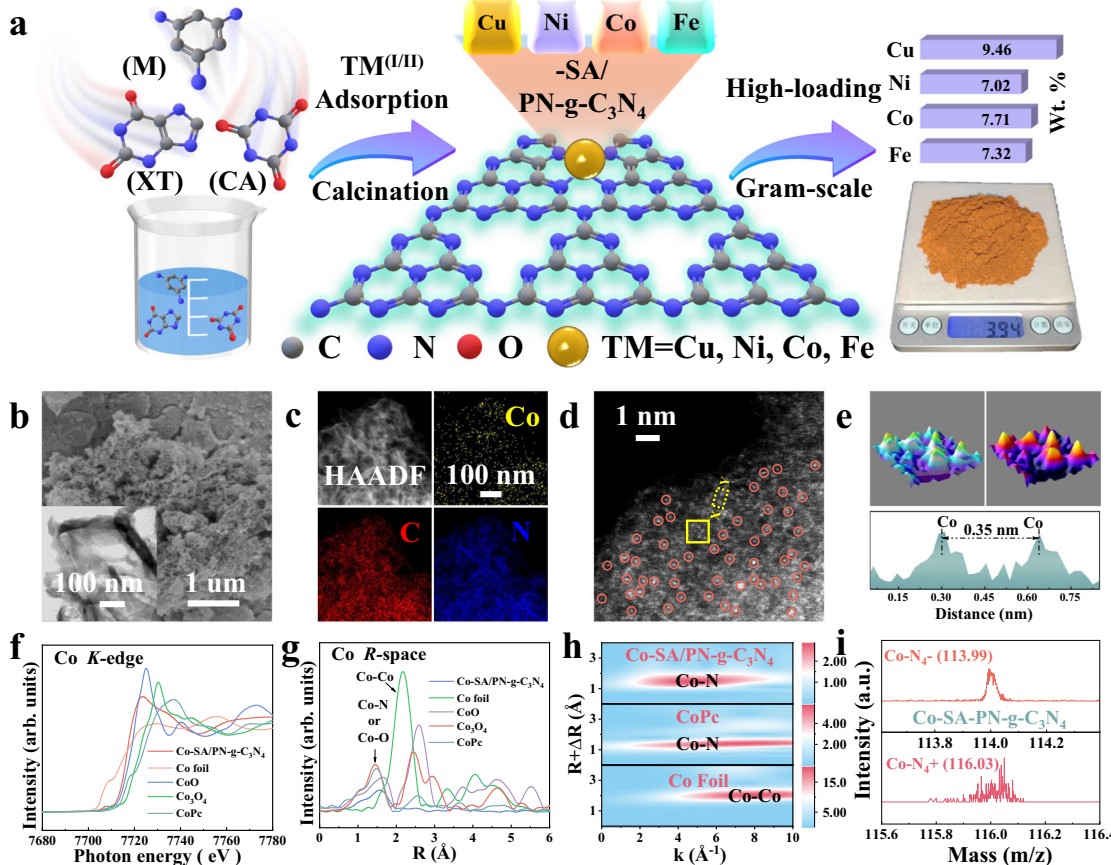

**Fig. 1 | Synthesis and characterization of the TM (Cu, Ni, Co, and Fe)-SA/PN-g-C₃N₄.** **a** Schematic illustration of the preparation procedures of the TM (Cu, Ni, Co, and Fe)-SA/PN-g-C₃N₄ samples. **b** SEM image (inset: HR-TEM image), (**c**) HAADF-STEM image with the corresponding EDS elemental mapping, and (**d**) AC HAADF-STEM image of Co-SA/PN-g-C₃N₄, the single atoms are marked with isolated red circles in (**d**). **e** 3D isolines and atom-overlapping Gaussian-function fitting mapping

of the yellow rectangle from (**d**) and intensity profile along $X–Y$ in (**d**). **f** Co $K$-edge XANES and (**g**) Fourier-transformed Co $R$-space EXAFS of Co-SA/PN-g-C₃N₄, Co foil, CoO, Co₃O₄, and CoPc. **h** WT-EXAFS plots of Co-SA/PN-g-C₃N₄, CoPc, and Co foil. **i** TOF-SIMS high-resolution ion spectra for Co−N₄ structural unit. Source data are provided as a Source Data file.

These characterizations show that the TM-SA/PN-g-C₃N₄ catalysts (TM = Cu, Ni, Co, and Fe) exhibited similar physicochemical properties and coordination structure (e.g., TM–N₄), which provided an ideal platform for revealing the electronic structure-dependent catalytic feature in SACs-AOPs.

## Pivotal roles of high-valent metal-oxo species in pollutant removal

The Fenton-like catalytic capabilities of the four TM-SA/PN-g-C₃N₄ catalysts were examined for phenol (PhOH) removal in the presence of PMS. The reduction in PhOH concentration was not attributed to adsorption by the PN-g-C₃N₄ substrate (Supplementary Fig. 13) or degradation catalyzed by the leached metal ions (Supplementary Fig. 14), suggesting the principal role of the embedded single metal species in PhOH removal. As shown in Fig. 2a, the observed reactivity sequence of the catalysts followed the order of Co > Fe >Ni > Cu, which is consistent with the order of turnover frequency values (TOF, mass-specific activity, Supplementary Fig. 15). Understanding the molecular origin of such a reactivity sequence is critical for highly efficient SAC fabrication for Fenton-like catalysis, but it faces a great challenge due to the contentious PMS activation mechanism[11]. Therefore, experiments were conducted to elucidate the reaction mechanism with electron paramagnetic resonance (EPR), reactive species quenching, isotope labeling, methyl phenyl sulfoxide (PMSO) trapping, in situ Raman spectroscopy, and soft X-ray absorption spectra (XAS). As shown in Fig. 2b, a typical seven-line EPR signal of

5,5-dimethyl-1-pyrrolidone-N-oxyl (DMPOX) was observed in the coexistence of Co-SA/PN-g-C₃N₄ and PMS, which could be attributed to 5,5-dimethyl-1-pyrroline N-oxide (DMPO) oxidation by high-valent metals[33] or excessive amount of •OH[34] or ¹O₂[18]. The reactive species responsible for DMPOX generation also contributed to PhOH removal because of the significantly depressed signal intensity of DMPOX when PhOH was introduced (Fig. 2b). Reactive species quenching experimental results show that methanol (MeOH) and tert-butanol (TBA) negligibly inhibited the removal efficiency of PhOH within 30 min (Supplementary Fig. 16), implying that neither SO₄•⁻ nor •OH was produced. This result is also validated by the EPR studies, which show an increase rather than decrease in DMPOX signal intensity after MeOH addition (Fig. 2b). Furthermore, the characteristic triplet EPR peaks indicated the emergence of ¹O₂, a commonly proposed reactive species in PMS-AOP (Supplementary Fig. 17). It is widely reported that deuterium oxide (D₂O) can enhance the oxidation efficiency of ¹O₂ because it prolongs the lifetime of ¹O₂ by approximately 18 times ($68 \pm 1\,\mu s$ in D₂O vs. $3.7 \pm 0.4\,\mu s$ in H₂O)[35,36]. However, the solvent exchange (H₂O to D₂O) did not enhance PhOH degradation, suggesting the negligible contribution of ¹O₂ (Supplementary Fig. 18). In addition, the contribution of O₂•⁻ and surface-attached radicals were also excluded by the EPR experiments[18,37] (Supplementary Figs. 19 and 20, and Supplementary Note 2). Thus, these results demonstrate that high-valent Co species, i.e., Co(IV)=O, might be primarily responsible for PhOH removal and DMPOX generation in the Co-SA/PN-g-C₃N₄/PMS system.

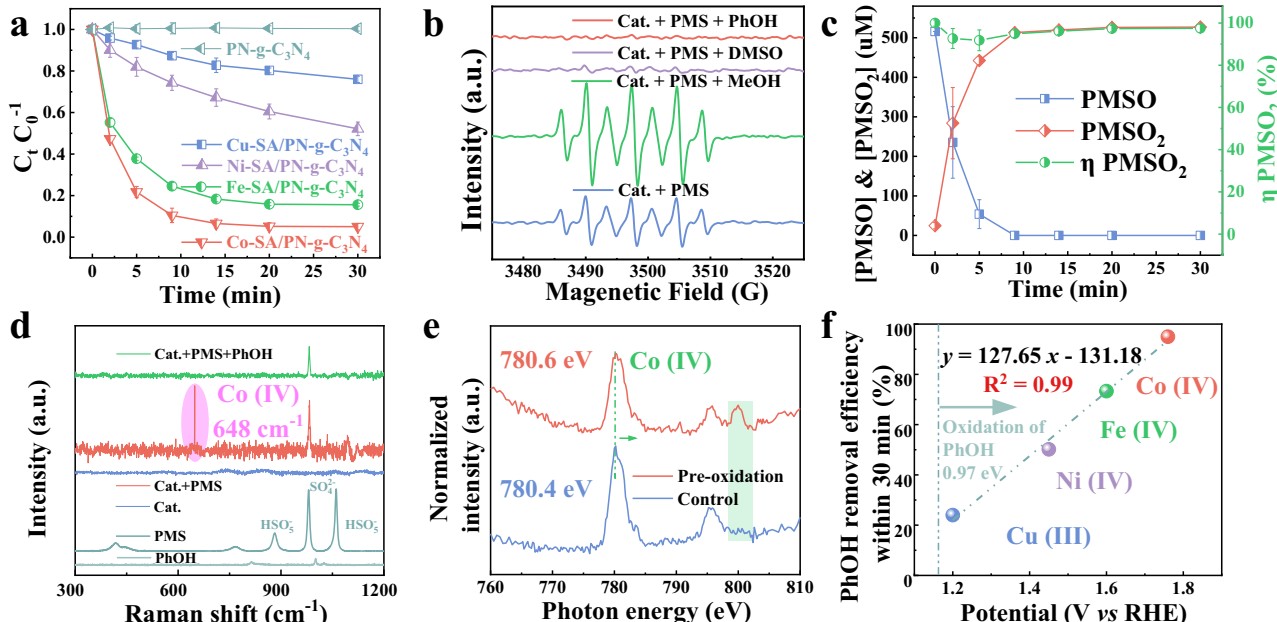

**Fig. 2 | The pivotal role of Co(IV)=O for pollutant removal. a** Comparison of pollutant removal efficiency between TM (Cu, Ni, Co, and Fe)-SA/PN-g-C₃N₄. Error bars represent the standard deviation, obtained by repeating the experiment three times. **b** EPR spectra of the different systems with DMPO as the spin-trapping agent. **c** PMSO probe experiment in Co-SA/PN-g-C₃N₄ system. Error bars represent the standard deviation, obtained by repeating the experiment two times. Reaction conditions in (**a**–**c**) [PMS] = 1.0 mM, [PhOH] = [PMSO] = 0.5 mM, [Cat.] = 1.0 g L⁻¹, [DMPO] = [MeOH] = 100 mM, [DMSO] = 10 mM, initial pH = 7.0, and $T = 25 \pm 2\,°C$. **d** In situ Raman study (the pink shade represents the signal of Co(IV)=O). **e** Soft-XAS analyzes (the green shade represents the signal of Co(IV)=O). **f** Fitting of the relationship between the oxygen reduction potential of high-valent metals and the removal efficiency of PhOH within 30 min. Source data are provided as a Source Data file.

To provide compelling evidence for the crucial role of Co(IV) =O, the quenching experiment using dimethyl sulfoxide (DMSO) was conducted. DMSO has been widely recognized as an effective inhibitor of high-valent metals through the oxygen transfer reaction[38]. The quenching experiment results in Supplementary Fig. 16 show that ca. 10 mM DMSO completely inhibited PhOH degradation. It should be noted that DMSO not only eliminated Co(IV)=O, but also directly reacted with PMS at a reaction rate constant of $18 \pm 0.7\ M^{-1}\ s^{-1}$[18]. However, the catalytic activation of PMS by Co-SA/ PN-g-C₃N₄ was still observed in the presence of DMSO, due to a significantly higher decomposition rate of PMS in the mixed Co-SA/ PN-g-C₃N₄/DMSO system compared to DMSO alone (Supplementary Fig. 21)[18]. These results support the critical role of Co(IV)=O in PhOH removal in the Co-SA/PN-g-C₃N₄/PMS system, which was further verified by the substantial decrease in the EPR peak intensity of DMPOX when DMSO was introduced (Fig. 2b). Furthermore, PMSO, which can be oxidized by Co(IV)=O via the oxygen transfer pathway to yield the specific methyl phenyl sulfone (PMSO₂) product[18], was applied as a chemical probe to confirm the presence of Co(IV)=O. As shown in Fig. 2c, the yield of PMSO₂, (η(PMSO₂), the molar ratio of PMSO₂ produced to PMSO lost, was found to be approximately 100% over the reaction time, validating Co(IV)=O as the primary oxidative species generated in the Co-SA/PN-g-C₃N₄/PMS system. The presence of Co(IV)=O was further confirmed by in situ Raman studies. The new peak at 648 cm⁻¹ after the introduction of PMS could be attributed to the stretching vibration of the Co(IV)=O structure, which completely disappeared when PhOH was present (Fig. 2d), indicating the critical role of Co(IV)=O in PhOH removal. Additionally, the occurrence of Co(IV)=O was verified by XAS analyzes. As shown in Fig. 2e, the absorption peak of the original Co-SA/PN-g-C₃N₄ at 780.4 eV positively shifted by 0.2 eV after PMS addition, demonstrating the oxidation of the surface Co species. Meanwhile, a new peak at approximately 800 eV further confirms the generation of Co(IV)=O[39].

The surface Co(IV)=O species can be preserved on the Co-SA/PN-g-C₃N₄ through pre-oxidation with PMS in the absence of pollutants. Such a pre-oxidized catalyst exhibited an impressive direct oxidative capacity for PhOH removal (~29.0%), significantly higher than the adsorption capacity of the fresh Co-SA/PN-g-C₃N₄ (<1%) (Supplementary Fig. 29). X-ray photoelectron spectroscopy (XPS) was used to track the change in the valence state of Co throughout the catalytic reaction process involving PMS activation and PhOH degradation (Supplementary Fig. 30c). The observed trend of initial increase followed by subsequent decrease provides clear evidence for the involvement of the high-valent Co species in PhOH degradation. Similar to other high-valent metal-based Fenton-like systems[18,40], electrophilic Co(IV)=O with high-selective features could effectively remove pollutants (e.g., PhOH and 2,6-M-PhOH) with electron-donating groups (−OH and −CH₃) only, but failed to remove benzoic acid with an electron-withdrawing group (−COOH) (Supplementary Figs. 31–33). Importantly, the dominant role of high-valent metals (e.g., Cu(III)−OH, Ni(IV) =O, and Fe(IV)=O) for PhOH removal was also confirmed in the other TM-SA/PN-g-C₃N₄ catalysts under the same conditions for the Co-SA/ PN-g-C₃N₄. Detailed results and discussions can be found in Supplementary Figs. 13–35 and Supplementary Note 2. The unique electronic structures of these high-valent metals significantly influenced their redox potentials, as revealed by the cyclic voltammetry (CV) tests (Supplementary Fig. 36), which showed a linear correlation between the removal efficiency of PhOH and potential ($R^2 = 0.990$, Fig. 2f), further demonstrating the pivotal role of high-valent metals for pollutant removal in the SACs-catalytic PMS systems.

## Polymerization-driven pollutant removal mechanisms

In the Co-SA/PN-g-C₃N₄/PMS system, the aqueous TOC was synchronously removed with PhOH at a high efficiency (~100%, Fig. 3a). Such unusual disappearance of aqueous TOC resulted in the accumulation of PhOH cross-linking polymers on the surface of catalysts, which could be directly identified by the matter-assisted laser desorption/

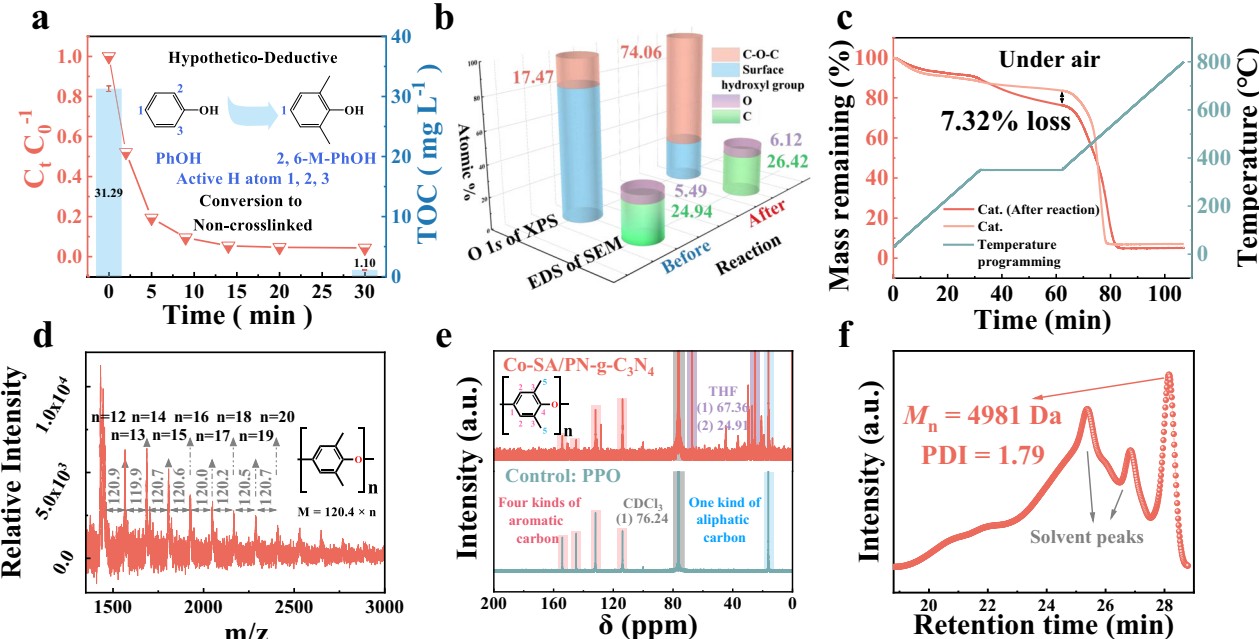

**Fig. 3 | Characteristics of the polymerization products of 2,6-M-PhOH on Co-SA/PN-g-C₃N₄. a** TOC and PhOH were removed synchronously and efficiently in Co-SA/PN-g-C₃N₄/PMS aqueous solution (Inset is schematic diagram of a hypothesis deductive method using 2,6-M-PhOH). The bar graphs correspond to the TOC values. Error bars represent the standard deviation, obtained by repeating the experiment three times. **b** SEM-EDS and O 1*s* of XPS, (**c**) TGA curves of Co-SA/PN-g-C₃N₄ before and after the reaction. **d** MALDI-TOF-MS, (**e**) NMR-based structural analysis (**e**), and (**f**) GPC of elution products on the surface of the Co-SA/PN-g-C₃N₄. The pink, blue, gray, and purple shades in **e** represent four kinds of aromatic carbon, one kind of aliphatic carbon, CDCl₃, and, THF, respectively. Reaction conditions in (**a–f**) [PMS] = 1.0 mM, [PhOH/2,6-M-PhOH] = 0.5 mM, [Cat.] = 1.0 g L⁻¹, initial pH = 7.0, and *T* = 25 ± 2 °C. Source data are provided as a Source Data file.

ionization time-of-flight mass spectrometry (MALDI-TOF-MS) tests (Supplementary Fig. 37). However, PhOH with three active hydrogen sites in its molecular structure (i.e., the ortho- and para-positions of the hydroxyl group) was readily oxidized to form a network-like cross-linked polymer, which was insoluble in organic solvents such as trihalomethanes (THMs), acetonitrile (ACN), and tetrahydrofuran (THF) (Supplementary Fig. 37a). Therefore, 2,6-dimethylphenol (2,6-M-PhOH) with only one active site, which tends to form non-crosslinked polymerization product, was selected to investigate the Co(IV)=O-initiated polymerization reactions (Fig. 3a), which were studied by multiple characterizations including SEM-EDS, XPS, and thermal gravimetric analysis (TGA). As shown in Fig. 3b and Supplementary Fig. 40, after the disappearance of 2,6-M-PhOH in solution, an increase in C and O elements on the Co-SA/PN-g-C₃N₄ was observed by SEM-EDS. This result was verified by the XPS O 1*s* results, which showed a distinct transition in surface O species from the hydroxyl group to C−O−C (Fig. 3b and Supplementary Figs. 43a and 44a). The post-reaction XPS C 1*s* results also reveal a significant increase in C=C/C−C and C−O bonds (Supplementary Figs. 45 and 46). In contrast, no considerable changes were noted in SEM-EDS or XPS spectra of the PN-g-C₃N₄ substrate before and after the reaction (Supplementary Figs. 42 and 46), signifying the crucial role of the metal center in pollutant polymerization. Also, negligible changes in XPS N 1*s* spectra suggest that the coordination environment of the single atom remained stable in the reaction (Supplementary Figs. 43b and 44b). These findings reveal a substantial accumulation of organic products on the catalyst surface. Such products could be pyrolyzed in the temperature programming process with 7.32% weight loss compared to the pristine Co-SA/PN-g-C₃N₄ as revealed by the TGA analyzes (Fig. 3c and Supplementary Figs. 47 and 48). To further understand the properties of the products formed on the Co-SA/PN-g-C₃N₄ surface, the deposited substances were separated using THF solvent and characterized by MALDI-TOF-MS, gel permeation chromatography (GPC), nuclear magnetic resonance (NMR), and Fourier transform

infrared spectroscopy (FTIR) analyzes. The eluted products, consisting of repeat units with an *m/z* of 120.4, exhibited a distinct polyphenylene oxide (PPO) structure, as evidenced by MALDI-TOF-MS (Fig. 3d). Additionally, the PPO structure of the polymerization product was also proven by the similar aromatic/aliphatic carbons and the aromatic C−O−C/C−H groups compared with the standard PPO sample, as revealed by the NMR and FTIR analyzes, respectively (Fig. 3e and Supplementary Fig. 51). The average *M*ₙ of the chain-like products was identified as 4981 Da (Fig. 3f).

Overall, the accumulated organics on the Co-SA/PN-g-C₃N₄ surface were characterized as PPO, which was also confirmed in the polymerization removal of 2,6-M-PhOH by the other TM (Cu, Ni, and Fe)-SA/PN-g-C₃N₄ catalysts (Supplementary Figs. 37–52 and Supplementary Note 3). The polymers on the catalyst surface would not affect the surface reactive sites, and they could be facilely collected by an elution-drying protocol with a recovery ratio of 81.57%, 81.30%, 88.43%, and 65.88% for Cu, Ni, Fe, and Co-SACs, respectively (Supplementary Figs. 53 and 54, and Supplementary Note 3). However, how the polymerization removal of pollutants was triggered by high-valent metal species is still elusive.

### Identification of phenoxyl radicals in polymer formation

Although phenoxyl radicals have been considered a key intermediate in polymer formation[13–15], whether phenoxyl radicals emerged and how to detect the low-activity radical in the Fenton-like polymerization systems remains challenging. Thus, two innovative protocols, in which (2,2,6,6-tetramethylpiperidin-1-yl) oxyl (TEMPO)-based traps containing alkyl (CHANT) were employed as the trapping agent and ferulic acid (FA) as the quenching agent, were applied to identify phenoxyl radicals in the PMS-AOPs.

Cross-coupling and spin-trapping are commonly used to stabilize transient radicals in EPR analysis (Supplementary Fig. 55a, b). However, these methods have deficiencies of limited structural information for trapped free radicals, poor sensitivity, and a high

incidence of false positives due to side reactions[41–43]. To overcome these drawbacks, we developed a detection protocol for phenoxyl radicals using a recently reported free radical trap, CHANT (Supplementary Fig. 56)[41]. In such a protocol, ultra-performance liquid chromatography-tandem mass spectrometry (UPLC-MS) was combined with EPR analysis to enable accurate and reliable detection (Supplementary Fig. 55c). As shown in Fig. 4a, the reaction of a phenoxyl radical with CHANT yielded a stable, non-radical product and released the TEMPO persistent radical, which were detected by UPLC-MS with $(M + H)^+/Z$ of 288.1930 (Fig. 4b) and probed by EPR with the typical triplet peaks (Fig. 4c), respectively. When DMSO, a scavenger for Co(IV)=O, was added, the EPR signal for the TEMPO persistent radical completely disappeared (Fig. 4c). Then, TEMPO, as a traditional cross-coupling agent, was also used to capture

phenoxyl radicals. The results show that phenoxyl radicals could cross-couple with TEMPO to form a stable adduct with $(M + H)^+/Z$ of 278.2075 (Supplementary Fig. 59d). In addition, inspired by the antioxidant function of FA in the field of phytochemistry[44], we developed FA as a phenoxyl radical scavenger to convert phenoxyl radicals back to the redox-inert phenol precursor (Fig. 4d). The results show that ca. 10 mM FA completely inhibited the removal of PhOH in the Co-SA/PN-g-C$_3$N$_4$/PMS system (Fig. 4e and Supplementary Fig. 61). Although FA can react with PMS directly, the catalytic activation of PMS by Co-SA/PN-g-C$_3$N$_4$ was still observed in the presence of FA (Supplementary Fig. 61f), suggesting the critical role of phenoxyl radicals in the polymerization reaction of pollutants. Thus, these results firmly support the Co(IV)=O-induced formation mechanism of phenoxyl radicals in the polymerization process of

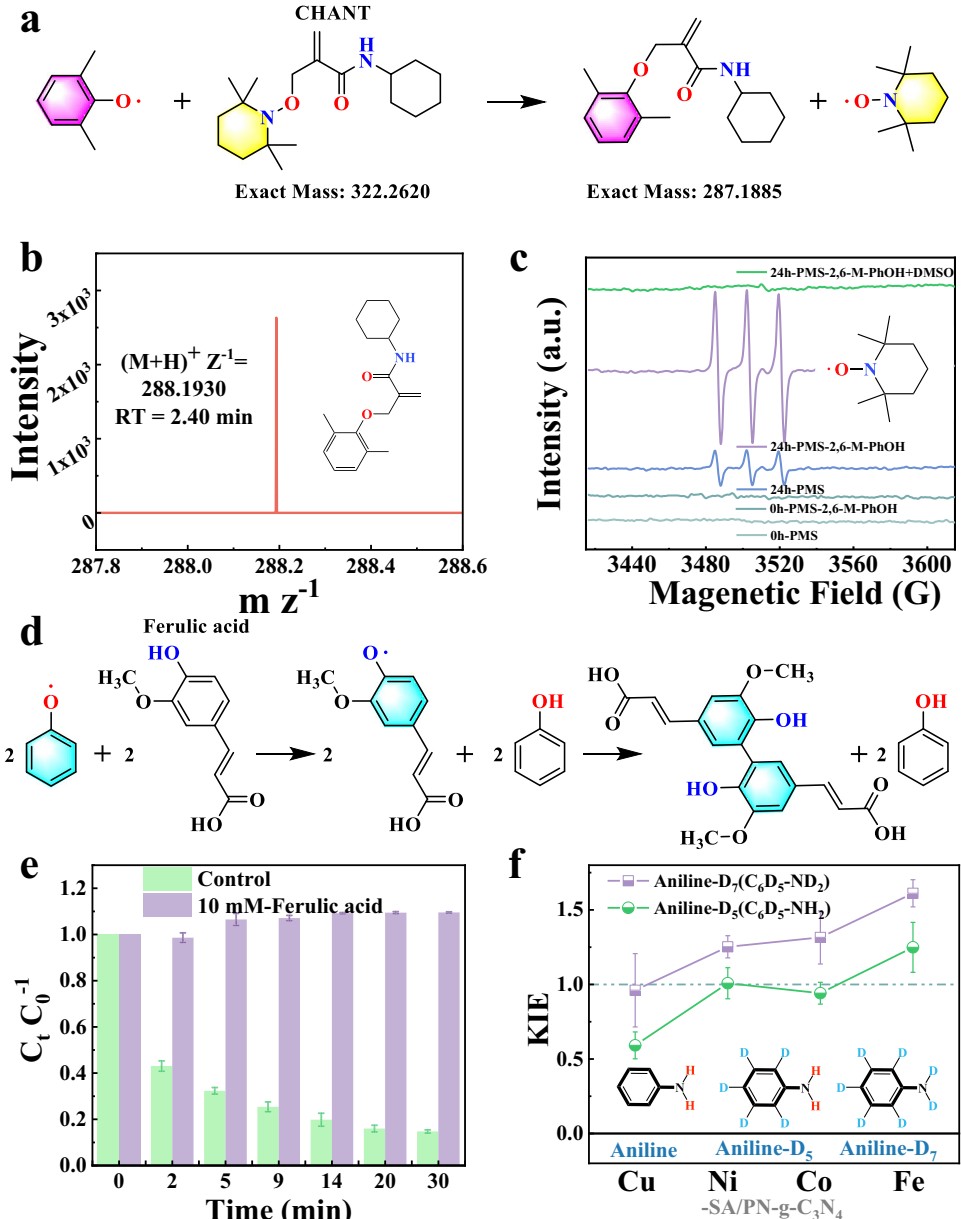

**Fig. 4 | Identification of phenoxyl radicals. a** A versatile approach for sequestering phenoxyl radicals using CHANT. **b** UPLC-MS chromatogram and (**c**) EPR spectra for probing the reaction product between CHANT and phenoxyl radicals. **d** Schematic illustration of phenoxyl radicals quenching by FA. **e** The effect of FA on PhOH degradation. Error bars represent the standard deviation, obtained by repeating the experiment two times. **f** H/D kinetic isotope effect for aniline

oxidation over the TM (Cu, Ni, Co, and Fe)-SA/PN-g-C$_3$N$_4$ system. Error bars represent the standard deviation, obtained by repeating the experiment two times. Reaction conditions: [Cat.] = 5.0 g L$^{-1}$, [PMS] = 10 mM, [2,6-M-PhOH] = [Aniline] = 5 mM, [CHANT] = 1 g L$^{-1}$, [DMSO] = [FA] = 10 mM, initial pH = 7.0, and $T = 25 \pm 2$ °C. Source data are provided as a Source Data file.

pollutants, the universality of such a mechanism was also confirmed for the other TM (Cu, Ni, and Fe)-SA/PN-g-C$_3$N$_4$ catalysts (Supplementary Figs. 55–61, Supplementary Note 4).

Moreover, it was reported that high-valent metals could oxidize pollutants into phenoxyl radicals via the hydrogen abstraction (HA) or proton-coupled electron transfer (PCET) pathway. The minor kinetic isotope effect (KIE) value excluded the HA pathway, suggesting that substrate oxidation was driven by PCET to generate the corresponding organic radicals in the four TM-SA/PN-g-C$_3$N$_4$/PMS systems (Fig. 4f and Supplementary Fig. 62). Overall, we discover the mediating role of the high-valent metals triggered phenoxyl radicals for the polymerization removal of pollutants. In addition, the method established for phenoxyl radical identification in this work is anticipated to have great application potential for a wide range of reactions involving organic free radicals, including photocatalysis, electrocatalysis, and organic synthesis[41,45–47].

## Regulating d-band center for polymerization removal of pollutants

High-valent metal species, encompassing Cu(III)−OH, Ni(IV)=O, Co(IV)=O, and Fe(IV)=O, are capable of initiating the polymerization removal of pollutants through the identical PCET pathway. However, these species also exhibit an electronic-structure-dependent behavior, leading to over-oxidation and decomposition of pollutants into small molecular organic acids.

Notably, the PT ratio for the Fe-SACs was substantially lower, at merely 44.9%, compared to the values of 99.3%, 98.0%, and 100.8% for the Cu, Ni, and Co-based SACs, respectively (Fig. 5a and Supplementary Fig. 32). This result suggests that pollutants could be completely polymerized on the surface of the TMs (Cu, Ni, and Co)-SA/PN-g-C$_3$N$_4$ catalysts. However, for the Fe-SA/PN-g-C$_3$N$_4$/PMS system, an additional pollutant removal mechanism, possibly involving the mineralization degradation pathway, might exist in this process. This hypothesis was further corroborated by the markedly lower $M_n$ of the chain-like polymer on the surface of the Fe-SA/PN-g-C$_3$N$_4$ catalyst, as indicated by the GPC analyzes (Fig. 5a and Supplementary Fig. 52). This finding implies the further degradation of pollutants into small molecule substances by the Fe(IV)=O species. Thus, understanding how the electronic structure of the catalytic center regulated the removal mechanism of the pollutants is essential to provide fundamental guidance for catalytic system design with impressive functions for pollutant polymerization removal from wastewater.

The distinct oxidative feature of Fe(IV)=O was revealed through a combination of density functional theory (DFT) calculations and experimental designs. As shown in Fig. 5b, the formation of phenoxyl radicals through the PCET mechanism includes three steps: adsorption of PMS on the TM-SA site (Step 1), the subsequent two-electron transfer leading to the production of high-valent metals (Step 2), and the oxidation of pollutant to yield phenoxyl radicals via the PCET pathway (Step 3). Eventually, the formed phenoxyl radicals would undergo further polymerization reaction on the catalyst surface. Thus, the electronic structure of the catalytic center could exhibit significant impacts on the pollutant removal mechanism by affecting the above-mentioned adsorption and electron transfer processes.

In the initial PMS adsorption process (Step 1), the projected density of states (PDOS) analyzes reveal that Fe site possesses the highest d-band center compared to that of Cu, Ni, and Co. Also, the d-band center of Fe further rises and approaches the Fermi-level ($E_F$) after PMS adsorption (Fe(II)-PMS) to form the high-valent metal-oxo (Fe(IV)=O). In contrast, Co and Ni sites exhibit decreasing trends throughout the aforementioned processes (Fig. 5c, d). Typically, a d-band center proximity to the $E_F$ generally indicates an enhanced propensity of the catalyst to donate charge to the adsorbate[48,49], and thus a precisely linear correlation between the d-band center of metals and the adsorption energy of PMS was observed (Fig. 5e and

Supplementary Fig. 65). Besides, the asymmetric arrangement of d-orbital electrons in the spin channel (5.79$_{(Fe)}$ > 1.76$_{(Co)}$) further validates the stronger bond strength and larger electron transfer due to spin polarization (Supplementary Fig. 67)[50]. When compared with the Co(IV)=O species, Fe(IV)=O conspicuously exhibits an augmented overlap of the Fe 3d and O 2p orbitals (3.630 vs. 2.487, Fig. 5f), leading to a more extensive electron delocalization over the Fe(IV)=O configuration and a stronger binding ability of the axial Fe for PMS[40]. The high affinity of Fe site for PMS was verified by the in situ Raman characterizations [a new peak at 840 cm$^{-1}$, corresponding to the activated peroxo-species (PMS*)], which was observed on the Fe-SA/PN-g-C$_3$N$_4$ only (Fig. 5g). Moreover, the O−O characteristic vibration peak of PMS* underwent a blue-shift of ~38 cm$^{-1}$ (from 1060 to 1098 cm$^{-1}$), indicating a strong interaction between PMS and the Fe-N$_4$ site on the catalyst[51]. Such peaks remained absent in the other TM-SA/PN-g-C$_3$N$_4$ catalyst spectra. Upon addition of pollutants, the simultaneous disappearance of signals corresponding to Fe(IV)=O and Fe-PMS* emphasized their collaborative role in pollutant elimination, operating through both PT and mineralization pathways (Supplementary Fig. 27).

To understand the configuration of PMS* on the catalyst surface, DFT calculations were carried out. Our computational insights reveal a marked affinity of PMS towards the Fe(IV) site in an axial orientation, culminating in the formation of the PMS*-Fe(IV)=O complex, a more favored configuration over the Fe(IV)=O-PMS* (Supplementary Fig. 70). At the subsequent stage, the pollutants tended to occupy the available O site. Notably, the adsorption energy linked with the PMS*-Fe(IV)=O complex was discernibly greater than that associated with PMS*-Co(IV)=O, as exemplified by the energy values of −0.292 eV for Fe and −0.244 eV for Co (Fig. 5h). Such an elevated adsorption energy typically resonated with an increased likelihood of undergoing oxidation. The validity of this assertion was underpinned by the observable substantial electron depletion in pollutants when in the proximity of PMS*-Fe(IV)=O (Fig. 5h and Supplementary Note 6).

Overall, the above simulation and experimental results demonstrate that the higher d-band center of Fe would result in a stronger interaction between Fe 3d and O 2p orbitals to form PMS*-Fe(IV)=O in the axial direction, which might subsequently initiate the mineralization of pollutants in the Fe-SA/PN-g-C$_3$N$_4$/PMS system.

For the oxidative generation of phenoxyl radicals (Step 3), the surface adsorption of PMS was observed to elongate the O-O bond with PMS, thus enhancing the oxidative capacity of the TM-SA/PN-g-C$_3$N$_4$/PMS systems when high-valent metals were involved. The different oxidative stresses significantly affected the subsequent behaviors of the phenoxyl radicals. To empirically ascertain this oxidative potential alteration (ΔV) post the creation of PMS*, open-circuit voltage measurements were conducted[23,52]. As shown in Fig. 6a, the positive shifts in open-circuit voltage upon introducing PMS indicate the elevation of the oxidative capacity across all four TM-SA/PN-g-C$_3$N$_4$/PMS systems. In a decreasing order of their amplitude, the voltage increments were arranged as: Fe > Cu > Ni > Co. Apparently, Fe with the highest d-band center and highest affinity for PMS showed the most significant increase in the oxidative capacity after PMS* formation (ΔV = 405 mV, Fig. 6a and Supplementary Fig. 75). The elevated oxidation capacity of the Fe-SA/PN-g-C$_3$N$_4$/PMS system after the Fe-PMS* complex formation was verified by various electrochemical characterizations. The chronoamperometric analyzes elucidated a swift dip in current trends post PMS infusion, implying an immediate electron transference from PMS to the metallic centers of the TM (comprising Cu, Ni, Co, and Fe) in the SA/PN-g-C$_3$N$_4$ catalysts, fostering the birth of high-valent metal entities (Fig. 6b). It is worth noting that when 2,6-M-PhOH was subsequently added, the current response initially dropped, but gradually increased for the Fe-SA/PN-g-C$_3$N$_4$, while it merely displayed a downtrend for the Cu-SA/PN-g-C$_3$N$_4$ (Fig. 6b). Such a vivid difference was a testament to the formidable oxidative caliber of the Fe-PMS* complex, which could robustly extract

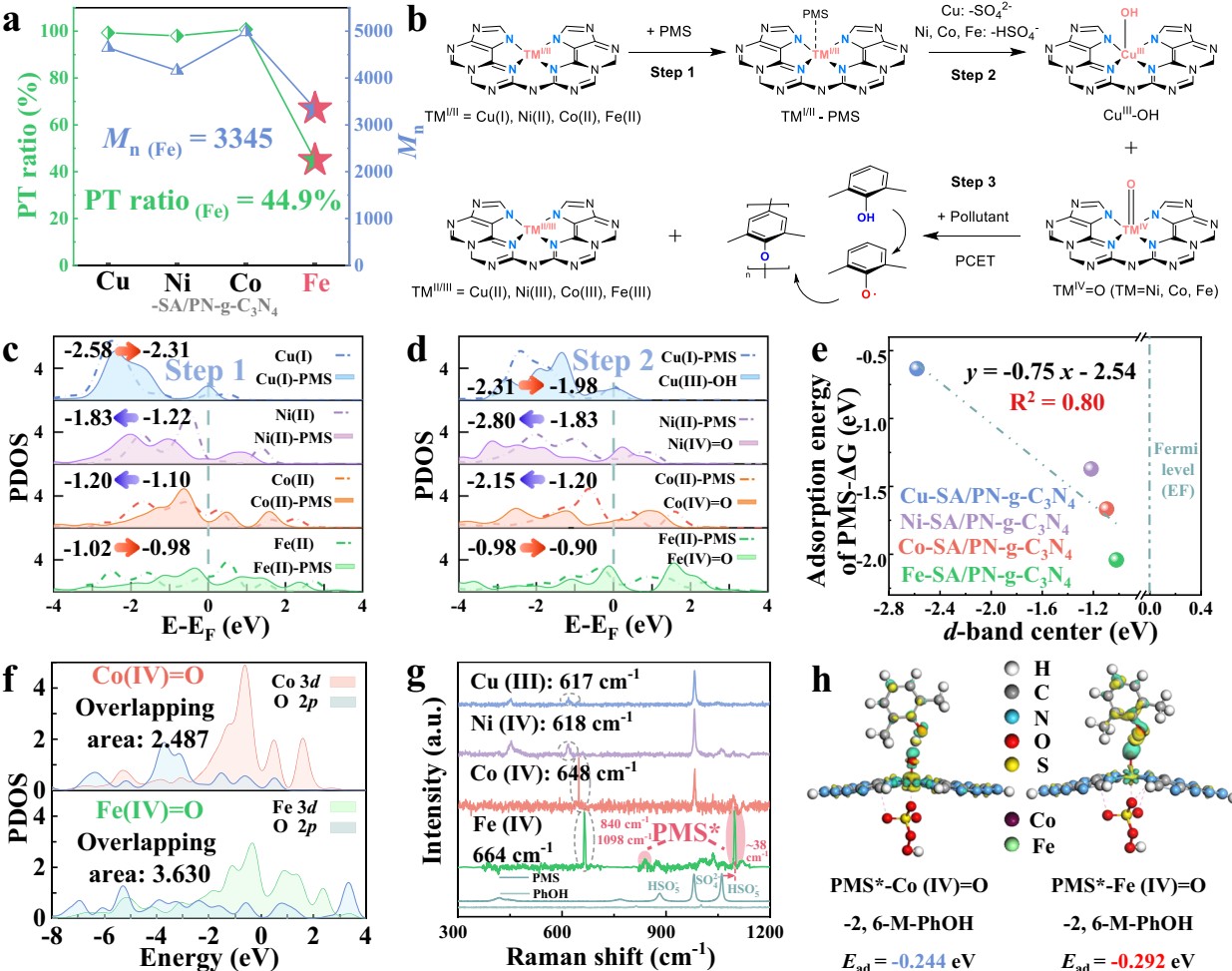

**Fig. 5 | Depicting the roles of the *d*-band center in oxidant and pollutant bindings. a** The PT ratios and $M_n$ of the elution products in the four TM-SA/PN-g-C$_3$N$_4$/PMS systems. Reaction conditions in **a** [PMS] = 1.0 mM, [PhOH] = 0.5 mM, [Cat.] = 1.0 g L$^{-1}$, initial pH = 7.0, and $T$ = 25 ± 2 °C. **b** Schematic diagram of phenoxyl radicals formation. PDOS of TM (Cu, Ni, Co, and Fe) $3d$ in TM-SA/PN-g-C$_3$N$_4$ systems of (**c**) step 1 and (**d**) step 2. **e** The relationship between the *d*-band center and the adsorption energy of PMS. **f** The PDOS of O $2p$ and TM (Fe and Co) $3d$ states on

TM(IV)=O structure, and the integrated overlapping area is labeled in the top left. **g** The identification of surface Fe-PMS* species (marked in red shade) by the Raman spectra. **h** Charge density difference and corresponding theoretical adsorption energy ($\Delta E_{ads}$) for PMS adsorption in the axial direction, in which light green and yellow areas indicate electron accumulation and depletion regions, respectively (isosurface = 0.03 e bohr$^{-3}$). Source data are provided as a Source Data file.

electrons from 2,6-M-PhOH, manifested as an enhanced current density uptick[53]. To quantitatively describe the oxidation capacity of the TM-SA/PN-g-C$_3$N$_4$/PMS system, $E_{system-TM}$, which represents the sum of the redox potential of the high-valent metals (Fig. 2f) and the increment in open circuit voltage after PMS addition ($\Delta V$, Fig. 6a), was therefore defined (Eq. 2). The $E_{system-TM}$ values were determined to be 2.01, 1.77, 1.48, and 1.37 V for the Fe, Co, Ni, and Cu SAC-based reaction systems, respectively.

$$E_{system-TM(Cu,Ni,Co,Fe)} = E_{TM^{(n+2)}} + \Delta V_{Open-circuit(TM-PMS^*)} \qquad (2)$$

As shown in the Gibbs free energy change plot (Fig. 6c), a spontaneous thermodynamic progression for phenoxyl radical formation could be observed for the four TM-SA/PN-g-C$_3$N$_4$/PMS systems with the tendency in the order of Fe > Co > Ni > Cu. The linear relationship between the $E_{system-TM}$ and the Gibbs free energy for phenoxyl radicals formation demonstrated phenoxyl radicals as the key intermediate in removing phenolic pollutants in the TM-SA/PN-g-C$_3$N$_4$/PMS systems (Fig. 6d). Further elucidation from UPLC-MS analyzes provides a transformative trajectory of 2,6-M-PhOH in tandem with the augmentation of the oxidative capacity inherent to the

TM-SA/PN-g-C$_3$N$_4$/PMS system. Molecular transformations encompassing hydroxylation, ring cleavage, and subsequent derivation of elementary organic acids such as glycolic, pyruvic, and mesoxalic acid were discerned within the Cu/Ni, Co, and Fe-SA/PN-g-C$_3$N$_4$/PMS systems, respectively (Fig. 6e and Supplementary Figs. 76–80). These findings substantiate that, beyond the well-documented polymerization pathway, the Fe-SA/PN-g-C$_3$N$_4$/PMS system with the paramount oxidation capacity can induce the over-oxidation of pollutants. This process culminates in the generation of organic acids that persist in the bulk solution, leading to the most diminished PT ratio.

Therefore, tailoring *d*-band center of high-valent metal-oxo species can achieve effective pollutant removal via complete polymerization, which exhibits impressive robustness against varying environmental conditions. The catalytic activities of the Cu, Ni, and Co-SA/PN-g-C$_3$N$_4$ catalysts, which feature a polymerization catalytic pathway, were negligibly affected by the co-existed anions (e.g., Cl$^-$, NO$_3^-$, SO$_4^{2-}$, and HCO$_3^-$) and water matrices (e.g., tap water, lake water, and secondary effluent from wastewater treatment plants). This result underscores the robustness of these catalysts in diverse aqueous environments. In contrast, the catalytic activity of Fe-SA/PN-g-C$_3$N$_4$ was slightly inhibited due to the involvement of a higher proportion of

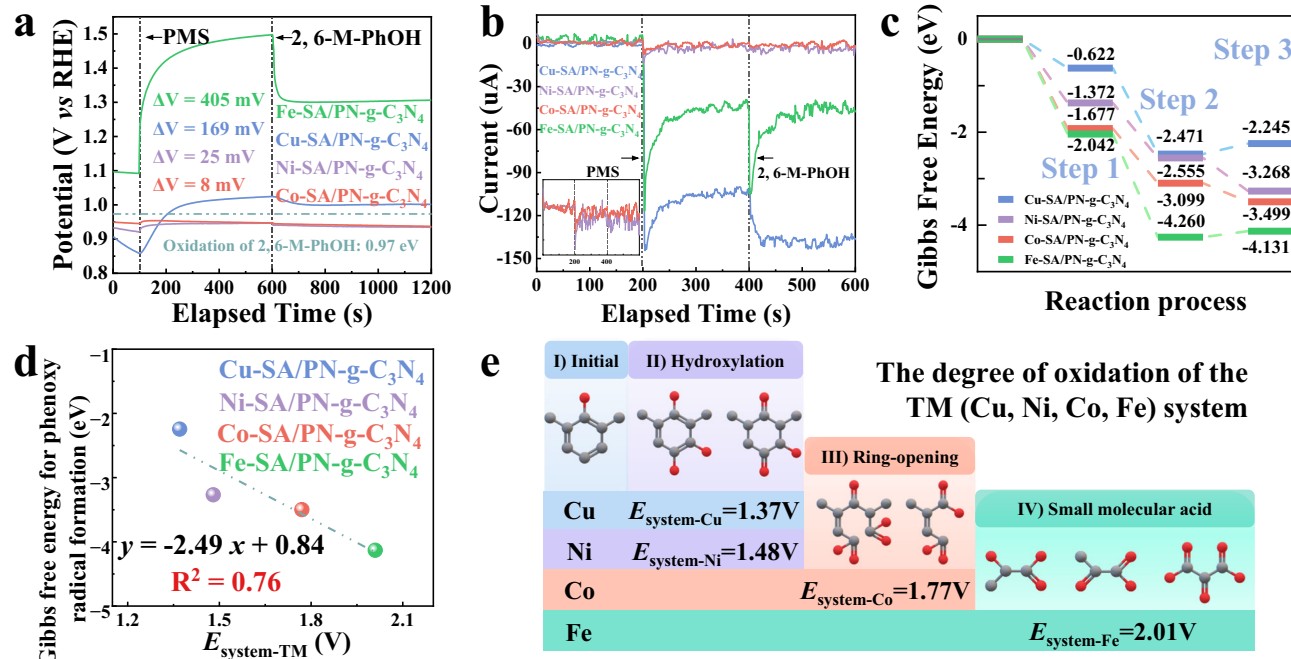

**Fig. 6 | The origin of the over-oxidation of pollutants in the Fe-SA/PN-g-C₃N₄/ PMS system. a** Open-circuit voltages after the sequential injection of PMS and 2,6-M-PhOH at the TM (Cu, Fe, Ni, and Co)-SA/PN-g-C₃N₄ working electrodes. **b** Chronocurrent diagrams for the sequential addition of PMS and 2,6-M-PhOH. **c** Gibbs free energy profiles of the three reaction steps for PMS adsorption, high-valent metal-oxo species formation, and phenoxyl radical formation in the different TM (Cu, Fe, Ni, and Co)-SA/PN-g-C₃N₄/PMS systems. **d** The relationship between the oxidation capacity of the reaction system and the Gibbs free energy for phenoxyl radical formation. **e** The degree of oxidation in the different TM (Cu, Fe, Ni, and Co)-SA/PN-g-C₃N₄/PMS systems. Source data are provided as a Source Data file.

mineralization degradation (-55.1%). In addition, an aqueous pH ranging from 4.0 to 9.0 had little effect on the removal efficiency or the PT ratio of PhOH in the four TM-SA/PN-g-C₃N₄/PMS systems. These results demonstrate the impressive anti-interference ability of the polymerization reaction in real wastewater treatment, confirming its versatility and effectiveness in pollutant removal (Supplementary Figs. 81–86 and Supplementary Note 5).

## Discussion

In summary, our work provides fundamental insights into the regulation of the catalytic polymerization and mineralization pathways for pollutant removal through modulating the *d*-band center of the TM (TM: Cu, Ni, Co, and Fe)-SACs (Supplementary Fig. 87). Both DFT calculations and experimental investigations prove that the high-valent metal-oxo species (e.g., Cu(III)−OH, Ni(IV)=O, and Co(IV)=O) with lower *d*-band centers could achieve nearly 100% PT ratio for pollutant removal, whereas Fe(IV)=O with a higher *d*-band center could strengthen the re-binding of PMS, thus resulting in a higher oxidative capacity to over-oxidize pollutants into small molecular acids and with the lowest PT ratio (44.9%). In addition, the high-valent metal-oxo species could trigger the one-electron oxidation of pollutants to generate the key phenoxyl radicals intermediate for further polymerization reactions, which was identified by a combination of innovative spin-trapping and quenching methods. This work provides deep insights into the structure–function relationship between the electronic structure of the catalytic center and the PT ratio for pollutant removal at the atomic level. The results would be helpful for developing sustainable water purification technologies to simultaneously realize contamination abatement and resource recovery in a low-carbon manner.

## Methods
### Chemicals

Iron nitrate nonahydrate (Fe(NO₃)₃•9H₂O, 99.9%), copper nitrate trihydrate (Cu(NO₃)₂•3H₂O, 99%), cobalt nitrate hexahydrate

(Co(NO₃)₂•6H₂O, 99.99%), nickel nitrate hexahydrate (Ni(NO₃)₂•6H₂O, 98%), chromium nitrate hexahydrate (Cr(NO₃)₃•9H₂O, 99.0%), zinc nitrate hexahydrate (Zn(NO₃)₂•6H₂O, 99%), melamine (C₃H₆N₆, 99%), sodium thiosulfate (Na₂O₃S₂, 97%), potassium iodide (KI, 99%), MeOH (CH₄O, 99.5%), ethanol (EA, C₂H₆O, 99.5%), and TBA (C₄H₁₀O, 99.5%) were purchased from Sinopharm Chemical Reagent Co., China. PhOH (C₆H₆O, 99%), 2,6-M-PhOH (C₈H₁₀O, 99%), benzoic acid (C₇H₆O₂, 99.5%), xanthine (C₅H₄N₄O₂, 98%), cyanuric acid (C₃H₃N₃O₃, 98%), PMS (2KHSO₅•KHSO₄•K₂SO₄, ≥42%), FA (C₁₀H₁₀O₄, 99%), D₂O (99 atom% D), aniline (C₆H₇N, 99.5%), DMSO (C₂H₆OS, 99%), methyl phenyl sulfoxide (PMSO, 97%), methyl phenyl sulfone (PMSO₂, 97%), 2,2,6,6-tetramethyl-4-piperidinol (TEMP, 98%), and DMPO (97%) were acquired from Aladdin Chemistry Co., China. Carbon papers were purchased from Toray Co., Japan. Aniline-D5 (98 atom% D) and Anline-D7 (98 atom% D) were obtained from Shanghai Civi Chemical Technology Co., China and J&K Scientific Co., China, respectively. Nafion solution (5 wt%) and ACN (>99%) were bought from Sigma-Aldrich Co., China. Deionized (DI) water (R = 18.25 MΩ) was used in all the experiments. All chemicals were used as received without further purification.

### Catalyst preparation

For the synthesis of the TM-(TM: Cu, Ni, Co, and Fe) SA/PN-g-C₃N₄ catalysts, cyanuric acid (9.6 mM) and xanthine (2.4 mM) were first dissolved into 80 mL DI water under ultrasonication. Then, 80 mL of melamine solution (1.2 mM) was quickly mixed with the above solution to form a suspension, which was ultrasonically by 20 min and magnetically stirred for a further 4 h. Subsequently, the metal precursor solution was prepared by dissolving 9.6 mM of the corresponding metal salt in 40 mL DI water, which was added to the above suspension and stirred for 1 h to achieve the self-assembly of the metal-carbon polymer. The assemblies were centrifuged and washed with DI water several times and dried in a vacuum at 60 °C for 10 h. Finally, the dried samples were ground in a mortar and calcined at 550 °C under N₂ atmosphere with a heating rate of 5 °C min⁻¹ to obtain the TM-(TM = Cu, Fe, Ni, and Co) SA/PN-g-C₃N₄ catalysts. The PN-g-C₃N₄ substrate

was prepared according to the above procedures in the absence of metal precursors.

## Characterizations

The characterization methods including XRD, BET, XPS, SEM, HR-TEM, HAADF-STEM, FTIR, EPR, soft-XAS, and XAFS are described in detail in Supplementary Methods.

## Pollutant degradation experiments

Pollutant degradation experiments were conducted at least in duplicate in 50 mL beakers at $25 \pm 2\,^\circ\text{C}$ under magnetic stirring. In a typical test, a certain mass of TM-SA/PN-g-$C_3N_4$ was added to the aqueous solution containing the desired amounts of pollutant and PMS. Unless otherwise stated, the molar ratio of oxidants to pollutants was fixed at 2:1, and the initial pH was adjusted to 7.0 by 0.5 mM $H_2SO_4$ or NaOH. In the reaction process, 1 mL of the reaction suspension was taken out and quenched by $Na_2S_2O_3$, the mixture was then filtered through a 0.22 μm membrane for further analysis.

## Analytical methods

The qualitative and quantitative analytical methods including UPLC, UPLC-MS measurement, TOC measurements, quenching experiments, KIE Experiments, pre-oxidation experiments, MALDI-TOF measurement, C-NMR characterization, GPC measurement, and electrochemical tests are available in Supplementary Methods.

## Quantum chemical calculations

All DFT calculations were performed on CASTEP code[54], seeing Supplementary Methods for more details. The atomic coordinates of the optimized structure are provided with this paper.

## Data availability

The data that supports the findings of the study are included in the main text and supplementary information files. Raw data can be obtained from the corresponding author upon request. Source data are provided with this paper.

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

## Acknowledgements

This work was supported by the National Natural Science Foundation of China (22376072 to M.J.H., 52192684 to H.Q.Y., 51821006 to H.Q.Y., 22106159 to M.J.H., and 52027815 to H.Q.Y.), and the Students' Innovation and Entrepreneurship Foundation of University of Science and Technology of China (CY2022G11 to H.Z.L. and CY2023G020 to H.Z.L.). We thank the photoemission endstations BL1W1B in Beijing Synchrotron Radiation Facility (BSRF), BL14W1 in Shanghai Synchrotron Radiation Facility (SSRF), BL12B-a, BL11U, and BL10B in National Synchrotron Radiation Laboratory (NSRL), the Shiyanjia Lab (www.shiyanjia.com), and Dr. Mei Sun from the HRTEM group (Instrument center for Physical Science in University of Science and Technology of China) for help with characterizations. The numerical calculations were performed on the supercomputing system in the Supercomputing Center of University of Science and Technology of China.

## Author contributions

M.J.H., J.J.C., and H.Q.Y. conceived and designed the study and supervised the project. H.Z.L. carried out the synthesis, characterization, and catalytic performance test. X.X.S. and J.J.C. conducted the theoretical studies. B.B.W. and X.S.W. helped to synthesize CHANT trapping agent. H.L.L. helped to synthesize the catalysts. M.J.H., H.Z.L., J.J.C., and H.Q.Y. analyzed the results and co-wrote the paper. All authors contributed to discussion of the results and the manuscript.

## Competing interests

The authors declare no competing interests.
