## [Peer Review File · Nature Communications]

Tailoring α -band center of high-valent metal-oxo species for pollutant removal via complete polymerizationREVIEWER COMMENTS

Reviewer #1 (Remarks to the Author):

In this study, the authors presented an innovative strategy to remove organic pollutants from water via complete polymerization by regulating the d-band center of the high-valent metal-oxo species, supporting a low-carbon water purification approach. Such a regulation strategy provides a brand-new strategy for transforming the removal pathway of organic pollutants from mineralization to polymerization. In addition, this work also demonstrates phenoxy radicals as the key intermediate in polymerization reactions by spin-trapping and quenching approaches. The experiments and DFT calculations are generally well performed, and the meaningful results obtained in this work have great significance in advancing the water purification field. Therefore, I recommend the publication in Nat. Commun. after addressing the following concerns.

(1) The water purification processes that proceed through the polymerization reaction indeed have the potential to recover the organic resources in wastewaters, but such the reaction is highly selective, only pollutants with electron-donating groups can be polymerized and separated from the aqueous solution. Thus, the statement “making it impossible to recover ~20% of the chemical energy in sewage” in the introduction section should be more rigorous, it is recommended to suitable modify it.

(2) The lower actual oxidant consumption than the theoretical oxidant consumption in AOPs indicates a polymerization reaction feature, please explain it.

(3) The reactivity sequence of the four TM-SACs was in the order of Co>Fe>Ni>Cu, but we also noticed different metal loadings of Cu, Ni, Co, and Fe, which were 9.46, 7.02, 7.71, and 7.32 wt.%, respectively. What's reactivity sequence respects to the single atom for the four catalysts?

(4) The effect of the water matrix on the polymerization reaction should be included to verify the practical application potential of this innovative pollutant removal process.

(5) The font size and font color in Fig. 3 should be modified for a concise and clear understanding. In addition, the number “1” on the molecular structure in Fig. 3d is partially covered by the blue arrow. Please modify them.

(6) The experimental conditions should be supplemented and added in the figure legends accordingly, such as Figs.3 and 5.

Reviewer #2 (Remarks to the Author):

This work presents a significant contribution to the field of advanced oxidation processes (AOPs) for the removal of organic pollutants. It introduces an innovative approach by employing polymerization reactions, which are effectively controlled by adjusting the d-band center of high-valent metal-oxo species. This novel regulation strategy shifts the organic pollutant removal mechanism from the conventional mineralization process to polymerization, offering a fresh perspective in this research area. Moreover, the identification of phenoxy radicals as key reaction intermediates in AOPs, achieved through groundbreaking spin-trapping and quenching methods, marks a pivotal advancement. The

manuscript is well-articulated, with the reaction mechanisms being elucidated through a thorough and logical experimental framework. I recommend the publication of this work in Nature Communications, subject to minor revisions. Below are specific suggestions aimed at enhancing the quality of the manuscript.

1. The authors have skillfully shown that by selectively generating high-valent metal-oxo species, the polymerization-based removal of organic pollutants can be effectively controlled in SACs/PMS systems. It is noteworthy that the redox potential of these high-valent metal-oxo species is considerably lower than that of hydroxyl radicals, implying a more selective mechanism for pollutant removal. This raises an interesting question about the practical applications of this polymerization technology (PT). I am keen to understand the specific contexts or scenarios where PT would be most effective and applicable.
2. In the introduction section, the authors highlight the robust anti-interference capabilities of polymerization in AOPs. To further elucidate this point, it would be beneficial if the authors could demonstrate how common anions influence the removal efficiency and the polymerization ratio of phenol during the activation process of PMS. Specifically, a detailed analysis or empirical data delineating the influence of these anions on both the efficacy and the underlying mechanism of pollutant removal within the specified systems would significantly augment the comprehension of the polymerization process's robustness against varying environmental conditions.
3. Line 28, $\bullet\text{OH}$ should be " $\bullet\text{OH}$ ", Please check and ensure consistency in description throughout the manuscript.
4. In Fig.2f, the Co(IV) showed a high potential (RHE) and pollutant removal compared with Fe(IV). Whether it has some correlation with the information from Fig. 5a, 5e and 6e might be deserved to be considered and interpreted.
5. In Fig 3a, there is a noticeable color variation along the midpoint line. Is this indicative of something specific? If it doesn't hold any special significance, the figure may require revision for clarity.
6. In Fig. 4a and Supplementary Fig. 52, the trapping adducts in the four reaction systems are shown to have varying retention times. Could these adducts be identified as the same product across the different systems?

Reviewer #3 (Remarks to the Author):

General comments: This study delves into the origins of catalytic pollutant polymerization using a set of transition metal (Cu, Ni, Co, Fe) single-atom catalysts. It pinpoints the d-band center of the active site as the crucial factor driving the transfer of pollutants during polymerization. The identification of phenoxyl radicals is achieved through innovative spin-trapping and quenching approaches. By modulating the d-band center, the oxidation capacity of high-valent metal-oxo species can be easily adjusted, influencing their binding strength for peroxymonosulfate. This research introduces a novel paradigm for dynamically modulating the electronic structure of high-valent metal-oxo species and optimizing pollutant removal from wastewater through polymerization. However, some problems need to be solved before the publication.

1. Line 27-28 "The AOPs with a capacity of producing reactive species (e.g., $\text{SO}_4^{\bullet-}$ and $\bullet\text{OH}$) to

mineralize a vast array of pollutants are termed as M-AOPs.” Please unify the format of $\bullet\text{OH}$ and other radicals throughout the manuscript.

2. Line 65-66 “we present a systematic in-depth investigation into peroxymonosulfate (PMS)-based heterogeneous catalytic AOPs over a series of TM (Cu, Ni, Co, Fe)-SACs.” Please define TM when it first appeared.

3. Line 98-99 “Energy-dispersive spectroscopy (EDS) mapping images verify the uniform distributions of the metal, C, and N elements across these architectures (Fig. 1c, Supplementary Fig. 5). I cannot see anything in EDS of Ni and N in Supplementary Fig. 5b and Fe in Supplementary Fig. 5c. Please explain. The other pictures are also not very clear.

4. Line 107-108 “Furthermore, the intensity distribution along X-Y in Fig. 1d reveals the spacing of Co atoms to be approximately 0.35 nm.” How about the spacing for other metal atoms? I think this paragraph aimed to confirm the atomic loading of Cu, Ni, Co, and Fe on the TM-SA/PN-g-C₃N₄ samples without the existence of metal-derived crystalline structures, so I think you need to supplement the spacing value for other metal atoms.

5. Line 116 “...indicating that the valence state of the Co atom lay between Co⁰ and Co²⁺.” However, In Supplementary Fig. 27c, the original Co-SA/PN-g-C₃N₄ has Co²⁺ and Co³⁺. The two conclusions are contradictory, so necessary explanations should be given.

6. Line 175-176 “Reactive species quenching experimental results show that MeOH and TBA negligibly inhibited PhOH degradation (Supplementary Fig. 15), implying that neither SO₄^{•-} and $\bullet\text{OH}$ was produced.” I do not understand which catalyst you referred to here. Because in Co-SA/PN-g-C₃N₄, the inhibition of MeOH and TBA is not negligible, but obvious.

7. The Supplementary Fig. 16 is so confusing. In this picture, does the line with a light color mean (catalyst +TEMP), and the dark color line mean (catalyst +TEMP+PMS)? Why also put PhOH in the system of Co-SA/PN-g-C₃N₄? How about the (Co-SA/PN-g-C₃N₄ +TEMP)?

8. Line 184-185 “However, the solvent exchange (H₂O to D₂O) did not enhance PhOH degradation, suggesting the negligible contribution of $^1\text{O}_2$ ” The result does not correspond with the quenching experiments. In Supplementary Fig. 15, FFA has an obvious inhibition for all the systems. Please explain it.

9. Except for SO₄^{•-} and $\bullet\text{OH}$, you need to consider the effect of O₂^{•-} and the radicals attached on the catalyst surface.

10. Line 189-190 “To provide compelling evidence for the crucial role of Co(IV)=O, the quenching experiment using DMSO was conducted.” Why DMSO can be used to prove the role of Co(IV)=O? Please give the reason.

11. Line 219-220 “...the change in the valence state of Co throughout the catalytic reaction process involving 219 PMS activation and PhOH degradation (Supplementary Fig. 27d).” Figure 27d is for Fe-SA/PN-g-C₃N₄, not Co-SA/PN-g-C₃N₄.

12. Line 270-271 “These findings reveal a substantial accumulation of organic products on the catalyst surface.” When the polymer covers the catalyst, does it affect the exposure of the reactive site? How to regenerate the catalyst? How to collect this recyclable product? More discussions are needed.

Response to Reviewer 1's comments

In this study, the authors presented an innovative strategy to remove organic pollutants from water via complete polymerization by regulating the d-band center of the high-valent metal-oxo species, supporting a low-carbon water purification approach. Such a regulation strategy provides a brand-new strategy for transforming the removal pathway of organic pollutants from mineralization to polymerization. In addition, this work also demonstrates phenoxyl radicals as the key intermediate in polymerization reactions by spin-trapping and quenching approaches. The experiments and DFT calculations are generally well performed, and the meaningful results obtained in this work have great significance in advancing the water purification field. Therefore, I recommend the publication in Nat. Commun. after addressing the following concerns.

Reply: We express our deep appreciation for the time and efforts of the reviewer on our manuscript. Below, we have detailed our response and revisions.

1. The water purification processes that proceed through the polymerization reaction indeed have the potential to recover the organic resources in wastewaters, but such the reaction is highly selective, only pollutants with electron-donating groups can be polymerized and separated from the aqueous solution. Thus, the statement “making it impossible to recover ~20% of the chemical energy in sewage” in the introduction section should be more rigorous, it is recommended to suitable modify it.

Reply: Accepting the reviewer's suggestion, we have revised the descriptions in the revised manuscript as follows:

“Typically, in M-AOPs, the carbons in organic contaminants are eventually converted into CO₂, which is then released into the atmosphere. This process results in the inability to recover the substantial chemical energy contained within sewage” (page 3, lines 31-34).

2. The lower actual oxidant consumption than the theoretical oxidant consumption in AOPs indicates a polymerization reaction feature, please explain it.

Reply: The oxidative removal of organics in aqueous solution can proceed via either the mineralization pathway to form smaller-molecular-weight products or the polymerization pathway to form higher-molecular-weight products¹. For instance, to completely mineralize 1 mol of phenol (C₆H₅OH), a theoretical minimum of 28 mol of electrons is required. This is equivalent to 14 mol of PMS. However, the initiation of the polymerization pathway, characterized by the formation of phenoxyl radicals, requires 1 mol of electron only, or 0.5 mol PMS. These facts indicate that the consumption of oxidants in polymerization is significantly lower than in mineralization. This also implies that the lower actual oxidant consumption than the theoretical oxidant consumption in AOPs (mineralization) indicates a polymerization reaction feature.

3. The reactivity sequence of the four TM-SACs was in the order of Co>Fe>Ni>Cu,

but we also noticed different metal loadings of Cu, Ni, Co, and Fe, which were 9.46, 7.02, 7.71, and 7.32 wt.%, respectively. What's reactivity sequence respects to the single atom for the four catalysts?

Reply: To evaluate the mass-specific activity of the four TM-SACs, the turnover frequency (TOF) values were calculated using the second-order rate constant and the loading amounts of single atom (**Fig. R1**). The results show that the TOF sequence was consistent with the previously observed apparent catalytic activity sequence of Co>Fe>Ni>Cu. We have incorporated extensive discussion on these findings in both the revised manuscript and SI.

In the revised manuscript, lines 165-167:

“As shown in Fig. 2a, the observed reactivity sequence of the catalysts follows the order of Co>Fe>Ni>Cu, which is consistent with the order of turnover frequency values (TOF, mass-specific activity, Supplementary Fig. 15).”

In the revised SI, Lines 301-304, Supplementary Fig. 15:

“To elucidate the mass-specific activity of the four TM-SACs, the turnover frequency (TOF) values were calculated by using the second-order rate constant and the loading amounts of single atoms. The results show that the TOF sequence was consistent with the apparent catalytic activity sequence of Co>Fe>Ni>Cu.”

Fig. R1 | Kinetic analysis and TOF calculation for TM-SACs in PMS activation. a, The second-order kinetic fitting results. **b,** Calculation of TOF (mass specific activity). Reaction conditions: [Cat.] = 1.0 g L⁻¹, [PMS] = 1.0 mM, [PhOH] = 0.5 mM, initial pH = 7.0, T = 25 ± 2 °C.

This additional analysis not only reinforces our initial observations, but also offers a deeper insight into the intrinsic catalytic capabilities of each TM-SAC when taking their specific metal loadings into account.

4. The effect of the water matrix on the polymerization reaction should be included to verify the practical application potential of this innovative pollutant removal process.

Reply: We fully agree that assessing the impacts of different water matrices is essential to validate the practical applicability of our innovative pollutant removal process. Thus, the effect of different water matrices on polymerization reaction was

examined (**Fig. R2**). The results reveal that the catalytic activities of the Cu, Ni, and Co-SA/PN-g-C₃N₄ catalysts, which feature a polymerization catalytic pathway, were negligibly affected by the water matrices, including tap water, lake water, and secondary effluent from wastewater treatment plants (WWTPs). This observation underscores the robustness of these catalysts in diverse aqueous environments. In contrast, the catalytic activity of Fe-SA/PN-g-C₃N₄ was slightly inhibited due to the involvement of a higher proportion of mineralization degradation (~55.1%). Nevertheless, it is noteworthy that the polymerization transfer (PT) ratio in the Fe-SA/PN-g-C₃N₄/PMS system was insusceptible of the water matrices. These results demonstrate the outstanding anti-interference ability of the polymerization reaction in real wastewater treatment, confirming its versatility and effectiveness in pollutant removal.

To address the reviewer's concern, we have added the above figures and discussion to the revised manuscript (pages 21-22, lines 500-513) and SI (Supplementary Fig. 86).

Fig. R2 | Impact of water matrix on PhOH degradation and PT ratios in TM-SA/PN-g-C₃N₄/PMS systems. a-d, Effect of the water matrix on PhOH degradation in the four TM-SA/PN-g-C₃N₄/PMS systems. e-h, The corresponding PT ratios in the four reaction systems. Reaction conditions: [Cat.] = 1.0 g L⁻¹, [PMS] = 1.0 mM, [PhOH] = 0.5 mM, initial pH = 7.0, T = 25 ± 2 °C.

5. The font size and font color in Fig. 3 should be modified for a concise and clear understanding. In addition, the number “1” on the molecular structure in Fig. 3d is partially covered by the blue arrow. Please modify them.

Reply: Following the reviewer’s suggestion, we have made the modification in

the revised manuscript (Fig. 3, page 13, lines 302-310).

Fig. R3 | Characterization of the polymerization products of 2, 6-M-PhOH on Co-SA/PN-g-C₃N₄. **a**, TOC and PhOH were removed synchronously and efficiently in Co-SA/PN-g-C₃N₄/PMS aqueous solution (Inset is schematic diagram of a hypothesis deductive method using 2, 6-M-PhOH). **b**, **c**, SEM-EDS and O 1s of XPS (**b**), and TGA curves (**c**) of Co-SA/PN-g-C₃N₄ before and after reaction. **d**, **e**, **f**, MALDI-TOF-MS (**d**), NMR-based structural analysis (**e**), and GPC of elution products (**f**) on the surface of the Co-SA/PN-g-C₃N₄. Reaction conditions in (**a-f**): [PMS] = 1.0 mM, [PhOH/2, 6-M-PhOH] = 0.5 mM, [Cat.] = 1.0 g L⁻¹, initial pH = 7.0, T = 25 ± 2 °C.

6. The experimental conditions should be supplemented and added in the figure legends accordingly, such as Figs.3 and 5.

Reply: Accepting the reviewer's suggestion, we have added the experimental conditions to the legends of Figs. 3 and 5 in the revised manuscript.

In the revised manuscript, lines 304-310:

Fig. 3 | Characterization of the polymerization products of 2, 6-M-PhOH on Co-SA/PN-g-C₃N₄. **a**, TOC and PhOH were removed synchronously and efficiently in Co-SA/PN-g-C₃N₄/PMS aqueous solution (Inset is schematic diagram of a hypothesis deductive method using 2, 6-M-PhOH). **b**, **c**, SEM-EDS and O 1s of XPS (**b**), and TGA curves (**c**) of Co-SA/PN-g-C₃N₄ before and after reaction. **d**, **e**, **f**, MALDI-TOF-MS (**d**), NMR-based structural analysis (**e**), and GPC of elution products (**f**) on the surface of the Co-SA/PN-g-C₃N₄. Reaction conditions in (**a-f**): [PMS] = 1.0 mM, [PhOH/2, 6-M-PhOH] = 0.5 mM, [Cat.] = 1.0 g L⁻¹, initial pH = 7.0, T = 25 ± 2 °C.

In the revised manuscript, lines 444-454:

Fig. 5 | Depicting the roles of the d-band center in oxidant and pollutant bindings. **a**, The PT ratios and M_n of the elution products in the four TM-SA/PN-g-C₃N₄/PMS systems. Reaction conditions in (**a**): [PMS] = 1.0 mM, [PhOH] = 0.5 mM, [Cat.] = 1.0 g L⁻¹, initial pH = 7.0, T = 25 ± 2 °C. **b**, Schematic diagram of phenoxyl radicals formation. **c**, **d**, PDOS of TM (Cu, Ni, Co, Fe) 3d in TM-SA/PN-g-C₃N₄ systems of step

1 (c) and step 2 (d). e, The relationship between the d-band center and the adsorption energy of PMS. f, The PDOS of O 2p and TM (Fe, Co) 3d states on TM(IV)=O structure, and the integrated overlapping area is labeled in the top left. g, The identification of surface Fe-PMS* species by the Raman spectra. h, Charge density difference and corresponding theoretical adsorption energy (ΔE_{ads}) for PMS adsorption in the axial direction, in which light green and yellow areas indicate electron accumulation and depletion regions, respectively (isosurface = 0.03 e bohr⁻³).

Response to Reviewer 2's comments

This work presents a significant contribution to the field of advanced oxidation processes (AOPs) for the removal of organic pollutants. It introduces an innovative approach by employing polymerization reactions, which are effectively controlled by adjusting the d-band center of high-valent metal-oxo species. This novel regulation strategy shifts the organic pollutant removal mechanism from the conventional mineralization process to polymerization, offering a fresh perspective in this research area. Moreover, the identification of phenoxy radicals as key reaction intermediates in AOPs, achieved through groundbreaking spin-trapping and quenching methods, marks a pivotal advancement. The manuscript is well-articulated, with the reaction mechanisms being elucidated through a thorough and logical experimental framework. I recommend the publication of this work in Nature Communications, subject to minor revisions. Below are specific suggestions aimed at enhancing the quality of the manuscript.

Reply: We greatly appreciate the reviewer's valuable comments and suggestions on our manuscript. Our specific replies and revisions are listed as follows.

1. The authors have skillfully shown that by selectively generating high-valent metal-oxo species, the polymerization-based removal of organic pollutants can be effectively controlled in SACs/PMS systems. It is noteworthy that the redox potential of these high-valent metal-oxo species is considerably lower than that of hydroxyl radicals, implying a more selective mechanism for pollutant removal. This raises an interesting question about the practical applications of this polymerization technology (PT). I am keen to understand the specific contexts or scenarios where PT would be most effective and applicable.

Reply: The reactive species in the PT process were identified as high-valent metal-oxo species featuring high selectivity, which endorsed the PT process with several advantages over the conventional Fenton technology, including robust anti-interference capability, minimal sludge generation, and reduced oxidant consumption. Thus, in addition to common application scenarios such as secondary effluent treatment for municipal wastewater and industrial wastewater, the PT process exhibits distinct advantages in the following two scenarios:

(1) **Pollutant decontamination in high-salinity wastewater.** High-salinity wastewater presents a unique challenge due to its elevated total dissolved solids content, typically exceeding 1%. This type of wastewater often contains high concentration (up to 20,000 mg/L) of anions, especially Cl^- and SO_4^{2-} . The presence of these anions has a pronounced quenching effect on free radicals, thereby significantly impeding the effectiveness of conventional radical-based AOPs. In contrast, the PT process, which has an outstanding anti-interference capability, emerges as an ideal solution to address the challenges posed by high-salinity wastewater.

(2) **Selective removal of low-concentration emerging contaminants.** Due to the

interference of commonly present organic substances at high concentrations, traditional biological treatment processes, and radical-based AOPs can't effectively remove the emerging and low concentration ($\mu\text{g/L}$ - ng/L) contaminants. The PT process, distinguished by its high selectivity and outstanding anti-interference capability, stands out as an effective option for treating such low-concentration emerging contaminants.

These scenarios underscore the versatility and effectiveness of the PT process in a range of wastewater treatment contexts. Also, PT process can be integrated into the existing wastewater treatment systems to improve their efficiency, especially in scenarios where selective removal of stubborn or toxic organic compounds is challenging with the current technologies.

2. In the introduction section, the authors highlight the robust anti-interference capabilities of polymerization in AOPs. To further elucidate this point, it would be beneficial if the authors could demonstrate how common anions influence the removal efficiency and the polymerization ratio of phenol during the activation process of PMS. Specifically, a detailed analysis or empirical data delineating the influence of these anions on both the efficacy and the underlying mechanism of pollutant removal within the specified systems would significantly augment the comprehension of the polymerization process's robustness against varying environmental conditions.

Reply: To address the reviewer's concern about the robustness of the polymerization process against varying environmental conditions, we have conducted experiments to examine the effects of common anions and aqueous pH on the polymerization reaction (**Figs. R4-R8**). The results show that the catalytic activities of the Cu, Ni, and Co-SA/PN-g-C₃N₄ catalysts, which were featured with a polymerization catalytic pathway, could not be affected by the coexisting anions like Cl⁻, NO₃⁻, SO₄²⁻, and HCO₃⁻. This finding highlights the remarkable resistance of these catalysts to anionic interference. However, it was noted that the catalytic activity of the Fe-SA/PN-g-C₃N₄ catalyst could be slightly inhibited by HCO₃⁻ due to the involvement of a greater proportion of mineralization degradation (~55.1%). Nevertheless, the Fe-SA/PN-g-C₃N₄ catalyst remained largely unaffected by the presence of Cl⁻, NO₃⁻, and SO₄²⁻. In addition, an aqueous pH ranging from 4.0-9.0 had little effects on the removal efficiency or the PT ratio of PhOH in the four TM-SA/PN-g-C₃N₄/PMS systems. These results demonstrate the outstanding anti-interference ability of the polymerization reaction, underscoring its suitability for diverse wastewater treatment scenarios.

Accepting the reviewer's suggestion, we have added the related figures and discussion into the revised manuscript (pages 21-22, lines 500-513) and SI (Supplementary Figs. 81-85).

Fig. R4 | Impact of Cl^- concentration on phenol degradation and PT ratios in TM-SA/PN-g-C₃N₄/PMS systems. a-d, The effect of Cl^- concentration on PhOH degradation in the four TM-SA/PN-g-C₃N₄/PMS systems. **e-h**, The corresponding PT ratios in the four reaction systems. Reaction conditions: [Cat.] = 1.0 g L⁻¹, [PMS] = 1.0 mM, [PhOH] = 0.5 mM, initial pH = 7.0, T = 25 ± 2 °C.

Fig. R5 | Impact of NO_3^- concentration on phenol degradation and PT ratios in TM-SA/PN-g-C₃N₄/PMS systems. a-d, The effect of NO_3^- concentration on PhOH degradation in the four TM-SA/PN-g-C₃N₄/PMS systems. **e-h**, The corresponding PT ratios in the four reaction systems. Reaction conditions: [Cat.] = 1.0 g L⁻¹, [PMS] = 1.0 mM, [PhOH] = 0.5 mM, initial pH = 7.0, T = 25 ± 2 °C.

Fig. R6 | Impact of SO₄²⁻ concentration on phenol degradation and PT ratios in TM-SA/PN-g-C₃N₄/PMS systems. a-d, The effect of SO₄²⁻ concentration on PhOH degradation in the four TM-SA/PN-g-C₃N₄/PMS systems. e-h, The corresponding PT ratio in the four reaction systems. Reaction conditions: [Cat.] = 1.0 g L⁻¹, [PMS] = 1.0 mM, [PhOH] = 0.5 mM, initial pH = 7.0, T = 25 ± 2 °C.

Fig. R7 | Impact of HCO₃⁻ concentration on phenol degradation and PT ratios in TM-SA/PN-g-C₃N₄/PMS systems. a-d, The effect of HCO₃⁻ concentration on PhOH degradation in the four TM-SA/PN-g-C₃N₄/PMS systems. **e-h**, The corresponding PT ratio in the four reaction systems. Reaction conditions: [Cat.] = 1.0 g L⁻¹, [PMS] = 1.0 mM, [PhOH] = 0.5 mM, initial pH = 7.0, T = 25 ± 2 °C.

Fig. R8 | Impact of pH on phenol degradation and PT ratios in TM-SA/PN-g-C₃N₄/PMS systems. a-d, The effect of pH on PhOH degradation in the four TM-SA/PN-g-C₃N₄/PMS systems. **e-h,** The corresponding PT ratio in the four reaction systems. Reaction conditions: [Cat.] = 1.0 g L⁻¹, [PMS] = 1.0 mM, [PhOH] = 0.5 mM, initial pH = 7.0, T = 25 ± 2 °C.

3. Line 28, •OH should be "•OH ", Please check and ensure consistency in description throughout the manuscript.

Reply: We apologize for the mistake, which have corrected in the manuscript.

4. In Fig.2f, the Co(IV) showed a high potential (RHE) and pollutant removal compared with Fe(IV). Whether it has some correlation with the information from Fig. 5a, 5e and 6e might be deserved to be considered and interpreted.

Reply: In Fig. 2f, our result reveals a linear correlation between the PhOH removal efficiency and the redox potential of the high-valent metal ($R^2=0.990$), in which the Co(IV)=O species demonstrated a higher potential and pollutant removal efficiency compared to Fe(IV)=O. This trend can be elucidated using the *d*-band model, proposed by Bjørk Hammer and Jens Nørskov. This model provides insights into the interaction between adsorbates and transition metal surfaces². Therefore, the higher *d*-band center observed for Fe(IV)=O correlated with its enhanced axial adsorption of PMS (Fig. 5e). This interaction led to the formation of a PMS*-Fe(IV)=O complex, which exhibited a higher redox potential than PMS*-Co(IV)=O complex (2.01 V vs. 1.77 V), according to cyclic voltammetry (CV) tests and open circuit voltage measurements (Fig. 2f and Fig. 6a). The PMS*-Fe(IV)=O complex, characterized by its high redox potential, was likely to facilitate excessive oxidation of pollutants into small-molecule acids, thereby reducing the PT ratio (Figs. 5a and 6e). Thus, the distinctive redox behavior of these high-valent metal-oxo species plays a crucial role in influencing the efficiency and pathway of pollutant degradation, offering a deeper understanding of the catalytic mechanisms in these systems.

5. In Fig 3a, there is a noticeable color variation along the midpoint line. Is this indicative of something specific? If it doesn't hold any special significance, the figure may require revision for clarity.

Reply: Accepting the reviewer's suggestion, we have redrawn this figure (Fig. R9).

Fig. R9 | Characterizations of the polymerization products of 2, 6-M-PhOH on Co-SA/PN-g-C₃N₄. **a**, TOC and PhOH were removed synchronously and efficiently in Co-SA/PN-g-C₃N₄/PMS aqueous solution (Inset is schematic diagram of a hypothesis deductive method using 2, 6-M-PhOH). **b, c**, SEM-EDS and O 1s of XPS (**b**), and TGA curves (**c**) of Co-SA/PN-g-C₃N₄ before and after reaction. **d, e, f**, MALDI-TOF-MS (**d**), NMR-based structural analysis (**e**), and GPC of elution products (**f**) on the surface of the Co-SA/PN-g-C₃N₄. Reaction conditions in (**a-f**): [PMS] = 1.0 mM, [PhOH/2, 6-M-PhOH] = 0.5 mM, [Cat.] = 1.0 g L⁻¹, initial pH = 7.0, T = 25 ± 2 °C.

6. In Fig. 4a and Supplementary Fig. 52, the trapping adducts in the four reaction systems are shown to have varying retention times. Could these adducts be identified as the same product across the different systems?

Reply: We extend our gratitude to the reviewer for raising this pertinent question. Indeed, this query also emerged during our research process, prompting us to undertake verification to address it comprehensively (Supplementary Fig. 57 in the revised SI). The minor variations in retention times observed for the monitored substances could be attributed to their extremely low concentration, where even subtle differences in the matrix composition, possibly arising from the SAC samples, could lead to significant differences in retention times³.

To conclusively validate the identical trapping adducts in the four reaction systems, we have mixed the four postreaction systems and found that only one retention time signal was detected, which varied in the previous four TM-SACs systems (Supplementary Fig. 57 in the revised SI). This result confirms that the differences in retention signals did come from the influence of the sample matrix composition. The adduct in the mixture system had a mass-to-charge ratio (M+H)⁺/Z of 288.1929, which was completely consistent with the values in the individual reaction systems (288.1928, 288.1928, 288.1930, and 288.1929 for Cu, Ni, Co, and Fe-SACs, respectively). In summary, despite the different retention times across the four reaction systems, the trapping adducts were indeed the same substance.

Response to Reviewer 3's comments

General comments: This study delves into the origins of catalytic pollutant polymerization using a set of transition metal (Cu, Ni, Co, Fe) single-atom catalysts. It pinpoints the d-band center of the active site as the crucial factor driving the transfer of pollutants during polymerization. The identification of phenoxyl radicals is achieved through innovative spin-trapping and quenching approaches. By modulating the d-band center, the oxidation capacity of high-valent metal-oxo species can be easily adjusted, influencing their binding strength for peroxymonosulfate. This research introduces a novel paradigm for dynamically modulating the electronic structure of high-valent metal-oxo species and optimizing pollutant removal from wastewater through polymerization. However, some problems need to be solved before the publication.

Reply: We sincerely thank the reviewer for thoroughly examining our manuscript and providing very helpful comments to guide our revision. Our response to the reviewer's comments and revisions are listed below.

1. Line 27-28 "The AOPs with a capacity of producing reactive species (e.g., $\text{SO}_4^{\bullet-}$ and $\bullet\text{OH}$) to mineralize a vast array of pollutants are termed as M-AOPs." Please unify the format of $\bullet\text{OH}$ and other radicals throughout the manuscript.

Reply: Accepting the reviewer's suggestion, we have thoroughly reviewed the entire manuscript and ensured that the formatting of the hydroxyl radical ($\bullet\text{OH}$) and other radicals, including the sulfate radical ($\text{SO}_4^{\bullet-}$), is now consistent throughout the manuscript.

2. Line 65-66 "we present a systematic in-depth investigation into peroxymonosulfate (PMS)-based heterogeneous catalytic AOPs over a series of TM (Cu, Ni, Co, Fe)-SACs." Please define TM when it first appeared.

Reply: Accepting the reviewer's suggestion, we have added the definition of TM when it first appeared.

"Herein, we present a systematic in-depth investigation into peroxymonosulfate (PMS)-based heterogeneous catalytic AOPs over a series of transition metal (TM: Cu, Ni, Co, Fe)-SACs and reveal the underlying mechanism for achieving a 100% PT ratio by modulating the d-band center of the catalytic center." (Page 4, lines 65-68)

3. Line 98-99 "Energy-dispersive spectroscopy (EDS) mapping images verify the uniform distributions of the metal, C, and N elements across these architectures (Fig. 1c, Supplementary Fig. 5). I cannot see anything in EDS of Ni and N in Supplementary Fig. 5b and Fe in Supplementary Fig. 5c. Please explain. The other pictures are also not very clear.

Reply: The invisible Ni, Fe, and N elements in the EDS mapping were primarily due to the small pixel setting in the HAADF-STEM instrument in our initial analysis.

To rectify this, we have adjusted the pixel size of the element dots and re-exported these EDS mapping images (Figs. R10 and R11). Furthermore, we have replaced the less clear images in the manuscript (Fig. 1) and SI (Supplementary Fig. 5) with these updated, higher-resolution versions. These modifications ensure that the distribution of Ni, Fe, and N elements across the catalyst architectures is now clearly visible and can be accurately interpreted in the context of our work. Many thanks.

Fig. R10 | Synthesis and characterizations of the TM (Cu, Ni, Co, Fe)-SA/PN-g-C₃N₄. **a**, Schematic illustration of the preparation procedures of the TM (Cu, Ni, Co, Fe)-SA/PN-g-C₃N₄ samples. **b-d**, SEM image (**b**), inset is the HR-TEM image), HAADF-STEM image with the corresponding EDS elemental mapping (**c**), and AC HAADF-STEM image of Co-SA/PN-g-C₃N₄ (**d**). **e**, 3D isolines and atom-overlapping Gaussian-function fitting mapping of the square and intensity profile along *X*-*Y* in (**d**). **f**, **g**, Co *K*-edge XANES (**f**) and Fourier-transformed Co *R*-space EXAFS (**g**) of Co-SA/PN-g-C₃N₄, Co foil, CoO, Co₃O₄, and CoPc. **i**, WT-EXAFS plots of Co-SA/PN-g-C₃N₄, CoPc, and Co foil. **h**, TOF-SIMS high-resolution ion spectra for Co-N₄ structural unit.

Fig. R11 | HAADF-STEM and EDS elemental mapping of TM-SA/PN-g-C₃N₄ catalysts. a-c, HAADF-STEM images and corresponding EDS elemental mappings of (a) Cu-SA/PN-g-C₃N₄, (b) Ni-SA/PN-g-C₃N₄, and (c) Fe-SA/PN-g-C₃N₄.

4. Line 107-108 “Furthermore, the intensity distribution along X-Y in Fig. 1d reveals the spacing of Co atoms to be approximately 0.35 nm.” How about the spacing for other metal atoms? I think this paragraph aimed to confirm the atomic loading of Cu, Ni, Co, and Fe on the TM-SA/PN-g-C₃N₄ samples without the existence of metal-derived crystalline structures, so I think you need to supplement the spacing value for other metal atoms.

Reply: Accepting the reviewer’s suggestion, we have measured the atomic spacings in the TM (Cu, Ni, and Fe)-SA/PN-g-C₃N₄ samples (**Fig. R12**). The analysis reveals that the atomic spacings for Cu, Ni, and Fe-SACs were approximately 0.34, 0.36, and 0.35 nm, respectively, which are comparable to that of Co-SACs (0.35 nm). The related descriptions and figures have been added to the revised manuscript (page 6, lines 107-109) and SI (Supplementary Fig. 6) accordingly. This update ensures a more comprehensive and accurate representation of the atomic spacing characteristics in all the metal SACs studied, thereby addressing the reviewers’ concerns and enhancing the rigor of our work.

Fig. R12 | Aberration-corrected (AC) HAADF-STEM imaging and analysis of TM-SA/PN-g-C₃N₄ catalysts. a-c, Aberration-corrected HAADF-STEM images of (a) Cu-SA/PN-g-C₃N₄, (b) Ni-SA/PN-g-C₃N₄, and (c) Fe-SA/PN-g-C₃N₄. 1, 2, 3 represent corresponding 3D isolines, atom-overlapping Gaussian-function fitting mapping of the square, and intensity profile along X–Y, respectively.

5. Line 116 "...indicating that the valence state of the Co atom lay between Co^0 and Co^{2+} ." However, In Supplementary Fig. 27c, the original Co-SA/PN-g- C_3N_4 has Co^{2+} and Co^{3+} . The two conclusions are contradictory, so necessary explanations should be given.

Reply: Thanks a lot for the reviewer's suggestions. XANES and XPS provide insights into the oxidation states of elements, but they probe different aspects due to their varying penetration depths. XANES provides an average oxidation state of the atoms both in the bulk and on the surface, while XPS specifically targets the catalyst surface, typically within a 5-10 nm range⁴. Therefore, XANES show that the valence state of Co atom in Co-SA/PN-g- C_3N_4 lays between Co^0 and Co^{2+} , indicating a notable metal-support interaction within the Co-SACs sample as interpreted in the manuscript. The higher valence state of Co, as demonstrated by the XPS spectra (Supplementary Fig. 30c), might be attributed to oxidation by oxygen in the atmosphere.

6. Line 175-176 "Reactive species quenching experimental results show that MeOH and TBA negligibly inhibited PhOH degradation (Supplementary Fig. 15), implying that neither $\text{SO}_4^{\bullet-}$ and $\bullet\text{OH}$ was produced." I do not understand which catalyst you referred to here. Because in Co-SA/PN-g- C_3N_4 , the inhibition of MeOH and TBA is not negligible, but obvious.

Reply: It is true that in the Co-SA/PN-g- C_3N_4 /PMS system, MeOH and TBA had obvious inhibitory effects on the PhOH degradation rate, but they had negligible effects on the PhOH removal efficiency (approximately 5~10%) within 30 min. This result suggests that the primary mechanism of inhibition by these alcohols was likely due to the competitive adsorption at the catalyst surface, which retarded the reaction kinetics, rather than directly scavenging free radicals and thereby reducing the total amount of PhOH removed.

This interpretation is further substantiated by the disproportionately greater inhibitory effect of TBA compared to ethyl acetate (EA). In typical radical-based AOPs, EA, a scavenger of both $\text{SO}_4^{\bullet-}$ and $\bullet\text{OH}$, is expected to have a more pronounced inhibitory effect than TBA, which primarily scavenges $\bullet\text{OH}$ ⁵. However, the abnormally greater inhibitory effect of TBA in the Co-SA/PN-g- C_3N_4 /PMS system implies that $\text{SO}_4^{\bullet-}$ and $\bullet\text{OH}$ were not the main reactive species. Instead, this result could be attributed to the fact that TBA has a greater surface affinity for Co-SA/PN-g- C_3N_4 due to its higher dielectric constant compared to MeOH (12.47 vs 33.0)⁶. Thus, TBA with a higher surface affinity can inhibit the surface polymerization reaction.

To address the reviewer's concern, we have revised the related descriptions in the manuscript (page 8, lines 178-180), and added the above analyses into the revised SI (Supplementary Note 2, lines 756-765).

7. The Supplementary Fig. 16 is so confusing. In this picture, does the line with a light color mean (catalyst +TEMP), and the dark color line mean (catalyst +TEMP+PMS)? Why also put PhOH in the system of Co-SA/PN-g- C_3N_4 ? How about the (Co-SA/PN-

g-C₃N₄+TEMP)?

Reply: We apologize for the confusing arrangement of the figure in the previous submission. For the bottom two lines, the light-colored line indicates TEMP only, and the dark-colored line indicates (TEMP+PMS); no catalyst was added to these systems. These two lines were treated as the BLANK set. In addition, PhOH was added to the system to validate the contribution of ¹O₂ to PhOH degradation by monitoring the changing profiles in the EPR signal intensity.

To address the reviewer's concern, we have divided this figure into four individual sub-figures (Fig. R13) to display the occurrence of ¹O₂ in the four TM-SA/PN-g-C₃N₄/PMS systems (Supplementary Fig. 17).

Fig. R13 | a-d, EPR spectra of TM-SA/PN-g-C₃N₄/PMS/TEMP system. Reaction conditions: [Cat.] = 1.0 g L⁻¹, [PMS] = 1.0 mM, [PhOH] = 0.5 mM, [TEMP] = 10 mM, initial pH = 7.0, T = 25 ± 2 °C.

8. Line 184-185 “However, the solvent exchange (H₂O to D₂O) did not enhance PhOH degradation, suggesting the negligible contribution of ¹O₂” The result does not correspond with the quenching experiments. In Supplementary Fig. 15, FFA has an obvious inhibition for all the systems. Please explain it.

Reply: The impact of FFA on these reactions needs to be interpreted with a nuanced understanding of its multiple roles in the reaction system. In addition to its ability to scavenge ¹O₂, FFA can also inhibit PMS-based Fenton-like reactions through competitive adsorption, PMS quenching effect, and electron transfer inhibition⁷. Therefore, the inhibitory effect of FFA does not necessarily indicate the involvement of

$^1\text{O}_2$ in pollutant degradation. More reliable methods like solvent (D_2O) exchange and EPR experiments, are required to accurately assess the role of $^1\text{O}_2$ as detailed in SI (Supplementary Note 2). These results collectively demonstrate the negligible contribution of $^1\text{O}_2$ to pollutant removal in the TM-SA/PN-g- C_3N_4 /PMS systems.

9. Except for $\text{SO}_4^{\bullet-}$ and $\bullet\text{OH}$, you need to consider the effect of $\text{O}_2^{\bullet-}$ and the radicals attached on the catalyst surface.

Reply: To confirm the generation of $\text{O}_2^{\bullet-}$ in the four TM-SA/PN-g- C_3N_4 /PMS systems, we have conducted EPR trapping experiments in MeOH ⁸. The results show that a negligible sextet peak for $\text{O}_2^{\bullet-}$ was detected in these four reaction systems (Fig. R15), thus excluding the occurrence of $\text{O}_2^{\bullet-}$.

In addition, extra-added fluoride ions (F^-) with remarkable surface affinity can desorb surface radicals into aqueous solution⁹. However, the EPR signals for $\text{SO}_4^{\bullet-}$ and $\bullet\text{OH}$ were not observed in the filtrate (Fig. R16), suggesting that no surface-attached $\text{SO}_4^{\bullet-}$ or $\bullet\text{OH}$ was present in the four TM-SA/PN-g- C_3N_4 /PMS systems.

To address the reviewer's concern, we have added the related figures and descriptions to the revised manuscript (page 9, lines 188-190) and SI (Supplementary Figs. 19, 20 and Note 2).

Fig. R15 | a-d, EPR spectra of $\text{O}_2^{\bullet-}$ in TM-SA/PN-g- C_3N_4 /PMS/DMPO(MeOH) systems. Reaction conditions: [Cat.] = 1.0 g/L, [PMS] = 1.0 mM, [PhOH] = 0.5 mM, [DMPO] = 100 mM, initial pH = 7.0, $T = 25 \pm 2$ °C.

Fig. R16 | a-d, EPR spectra of TM-SA/PN-g-C₃N₄/PMS/DMPO/F⁻ systems. e, System control group. f-g, EPR spectra of filtrate in the TM-SA/PN-g-C₃N₄/PMS/DMPO/F⁻ systems. Reaction conditions: [Cat.] = 1.0 g L⁻¹, [PMS] = 1.0 mM, [PhOH] = 0.5 mM, [DMPO] = 100 mM, [NaF] = 10 mM, initial pH = 7.0, T = 25 ± 2 °C.

10. Line 189-190 “To provide compelling evidence for the crucial role of Co(IV)=O, the quenching experiment using DMSO was conducted.” Why DMSO can be used to prove the role of Co(IV)=O? Please give the reason.

Reply: DMSO is widely recognized as an effective inhibitor of high-valent metals via the oxygen transfer reaction^{8,10}. For instance, DMSO has a high reaction constant of $2.4 \times 10^6 \text{ M}^{-1} \text{ s}^{-1}$ for Co(IV)=O to form Co(II) and the oxygen transfer product DMSO₂ (Eq. 1)¹¹.

To address the reviewer’s concern, we have added the related description into the revised manuscript (page 9, lines 194-196).

11. Line 219-220 “...the change in the valence state of Co throughout the catalytic reaction process involving 219 PMS activation and PhOH degradation (Supplementary Fig. 27d).” Figure 27d is for Fe-SA/PN-g-C₃N₄, not Co-SA/PN-g-C₃N₄.

Reply: We apologize for this mistake, which have been corrected in the revised manuscript (page 10, line 227).

12. Line 270-271 “These findings reveal a substantial accumulation of organic products on the catalyst surface.” When the polymer covers the catalyst, does it affect the exposure of the reactive site? How to regenerate the catalyst? How to collect this recyclable product? More discussions are needed.

Reply: Thanks a lot for the valuable comments. To confirm whether the reactive sites of the catalysts are impacted by the polymer, we have conducted cyclic tests on the TM (Co and Fe)-SA/PN-g-C₃N₄ catalysts to assess the stability and performance of these catalysts in repeated PhOH degradation cycles. The results reveal a decrease in the PhOH removal efficiency of 12.95% and 4.43% after 5 consecutive reaction cycles for Co-SA/PN-g-C₃N₄ and Fe-SA/PN-g-C₃N₄, respectively, validating the superior stability of the catalysts for cyclic PhOH degradation (**Fig. R17a-b**). These results also suggest that the presence of polymers did not adversely affect the reactive sites on the catalyst surface, thus maintaining their catalytic efficiency.

To regenerate the catalysts, we applied two methods as depicted in **Fig. R17c**. For Method 1, tetrahydrofuran (THF) served as the solvent to dissolve and remove polymers accumulated on the catalyst surface. Method 2 involved the use of an atmospheric pyrolysis strategy, where the catalysts were subjected to heating at 350 °C for 30 min, effectively decomposing the surface-bound polymer. These two regeneration methods could completely recover the catalytic activities of the catalysts to their original levels, as evidenced in **Fig. R17a-b**.

The surface-accumulated polymers could be facilely collected by an elution-drying protocol. The postreaction catalysts were collected and repeatedly washed with THF to obtain a yellow solution; the supernatant was then dried in an oven at 80 °C to obtain the solid polymers. The recovery ratios for these polymers were quantified as 81.57%, 81.30%, 88.43%, and 65.88% for Cu, Ni, Fe, and Co-SACs, respectively (**Fig. R18**).

Accepting the reviewer’s suggestion, we have added the related descriptions and figures into the revised manuscript (page 13, lines 295-298) and SI (Supplementary Figs. 53-54 and Note 3).

Fig. R17 | Experimental procedures and conditions for cycling reuse and regeneration of the catalysts. a-b, Cycling reuse and regeneration experiments of TM (Co and Fe)-SA/PN-g-C₃N₄. **c,** two methods for regenerating the catalyst. Reaction conditions: [Cat.] = 1.0 g L⁻¹, [PMS] = 1.0 mM, [2, 6-M-PhOH] = 0.5 mM, initial pH = 7.0, T = 25 ± 2 °C.

Fig. R18 | Recovery of the polymers on the catalyst surface. **a**, Degradation kinetics of 2, 6-M-PhOH in the TM-SA/PN-g-C₃N₄/PMS systems. **b**, The corresponding TOC residuals and PT ratio values. **c**, The images of the collected polymers. **d**, Calculation of the recovery ratios. Reaction conditions: [Cat.] = 1.0 g L⁻¹, [PMS] = 4.0 mM, [2, 6-M-PhOH] = 2.0 mM, initial pH = 7.0, T = 25 ± 2 °C.

References

1. Zhang, Y. et al. Simultaneous nanocatalytic surface activation of pollutants and oxidants for highly efficient water decontamination. *Nat. Commun.* **13**, 3005 (2022).
2. Andersen, M. Revelations of the d band. *Nat. Catal.* **6**, 460-461 (2023).
3. Williams, M. L., Olomukoro, A. A., Emmons, R. V., Godage, N. H. & Gionfriddo, E. Matrix effects demystified: Strategies for resolving challenges in analytical separations of complex samples. *J. Sep. Sci.* **46**, (2023).
4. Anwar, M. et al. A review of x-ray photoelectron spectroscopy technique to analyze the stability and degradation mechanism of solid oxide fuel cell cathode materials. *Materials.* **15**, 2540 (2022).
5. Gu, C. et al. Slow-release synthesis of Cu single-atom catalysts with the optimized geometric structure and density of state distribution for Fenton-like catalysis. *Proc. Natl. Acad. Sci. U. S. A.* **120**, (2023).
6. Huang, M. et al. In situ-formed phenoxyl radical on the CuO surface triggers efficient persulfate activation for phenol degradation. *Environ. Sci. Technol.* **55**, 15361-15370 (2021).
7. Le Behec, M., Pigot, T. & Lacombe, S. Chemical quenching of singlet oxygen and other reactive oxygen species in water: a reliable method for the determination of quantum yields in photochemical processes? *Chemphotochem.* **2**, 622-631 (2018).
8. Huang, M. et al. Facilely tuning the intrinsic catalytic sites of the spinel oxide for peroxymonosulfate activation: from fundamental investigation to pilot-scale demonstration. *Proc. Natl. Acad. Sci. U. S. A.* **119**, e2092285177 (2022).
9. Wang, L. et al. A polymer tethering strategy to achieve high metal loading on catalysts for Fenton reactions. *Nat. Commun.* **14**, 7841 (2023).
10. Li, X. et al. CoN₁O₂ single-atom catalyst for efficient peroxymonosulfate activation and selective Cobalt(IV)=O Generation. *Angew. Chem. Int. Ed. Engl.* **62**, e202303267 (2023).
11. Zong, Y. et al. Unraveling the overlooked involvement of high-valent cobalt-oxo species generated from the cobalt(II)-activated peroxymonosulfate process. *Environ. Sci. Technol.* **54**, 16231-16239 (2020).

REVIEWERS' COMMENTS

Reviewer #1 (Remarks to the Author):

The manuscript has answered the questions we asked and is recommended for acceptance.

Reviewer #2 (Remarks to the Author):

The issues that i raised have been addressed properly, thus i recommend the publication in Nat. Commun.

Reviewer #3 (Remarks to the Author):

The comments have been addressed. OK to publish now.

Response to Reviewer 1's comments

The manuscript has answered the questions we asked and is recommended for acceptance.

Reply: We thank the reviewer for the assessment and support of our work for publication.

Response to Reviewer 2's comments

The issues that i raised have been addressed properly, thus i recommend the publication in Nat. Commun.

Reply: We really thank the reviewer for his/her positive comments.

Response to Reviewer 3's comments

The comments have been addressed. OK to publish now.

Reply: We thank the reviewer for the time and efforts on our work.